# Impact of decadal reversals of the North Ionian circulation on phytoplankton phenology

Héloise Lavigne[1], Giuseppe Civitarese[2], Miroslav Gacic[2], Fabrizio D'Ortenzio[3]

[1]Royal Belgian Institute of Natural Sciences, Operational Directorate Natural Environment, Gulledelle 100, B-1200 Brussels, Belgium.
[2]Istituto Nazionale di Oceanografia e di Geofisica Sperimentale – OGS, Dip. di Oceanografia, Borgo Grotta Gigante 42/c, 34010 Sgonico (Trieste), Italy
[3]Laboratoire d'Oceanographie de Villefranche, Université Pierre et Marie Curie et CNRS, UMR 7093, Villefranche-sur-Mer, France

*Correspondence to*: Héloïse Lavigne (hlavigne@naturalsciences.be)

**Abstract.** In the North Ionian, water circulation is characterized by a decadal alternation of cyclonic and anticyclonic regime driven by the mechanism called BiOS (Bimodal Oscillating System). The circulation regimes affect both vertical dynamics and the nutrient distribution. The North Ionian is then a good study area to investigate how changes in circulation can affect phytoplankton dynamics in oligotrophic regions. From in situ observations, for each circulation regime the averaged distribution of isopycnals is provided, and a depth difference of about 80m is estimated for the nitracline between cyclonic and anticyclonic regime. Based on phytoplankton phenology metrics extracted from annual time series of satellite ocean color data for the period 1998-2012, the cyclonic and anticyclonic regimes are compared. Results show that the average chlorophyll in March, the date of bloom onset and the date of maximum chlorophyll were affected by circulation patterns in the North Ionian. In the center of the North Ionian gyre, bloom onset occurred in December and chlorophyll was low in March when circulation was anticyclonic, whereas during the cyclonic circulation regime, a late chlorophyll peak, likely resulting from different phytoplankton dynamics, was commonly observed in March. An additional analysis shows that the winter buoyancy losses, which govern the Mixed Layer Depth (MLD) also contribute to explain the interannual variability in bloom onset and intensity. Two trophic regimes were then identified in the NIG and they could be explained with the relative position of the MLD and nitracline. The first one is characterised by an early winter bloom onset and the absence of chlorophyll peak in March. It was observed when circulation was anticyclonic or when winter MLD was relatively shallow. A dominant regenerated production all year round and an absence of significant nutrient supplies to surface waters are proposed to explain this trophic regime. Conversely, the second trophic regime is marked by a bloom onset in late winter (i.e. February) and a peak of chlorophyll in March. The chlorophyll increase was interpreted as the direct response to nutrient enrichment of surface waters. This winter/spring bloom was observed when circulation was cyclonic and when winter mixing was relatively strong.

# 1 Introduction

Phytoplankton phenology aims to analyze periodic events in phytoplankton seasonal cycles, such as blooms (Ji et al., 2010). Phytoplankton blooms are crucial events in the ocean as they contribute significantly to primary production and carbon export, (Buesseler, 1998) but also sustain the trophic food web. Thus, any changes in phytoplankton phenology may have large consequences on higher trophic levels (Edwards and Richardson, 2004; Koeller et al., 2009). Although the characterization of phytoplankton phenology in the open ocean has long been hindered by data scarcity, in the past few years, satellite ocean color observations have offered a general view of the phenology of the surface phytoplankton biomass at global scale (Platt et al., 2010). Satellite-based studies have mainly attempted to describe phytoplankton blooms (Yoder et al., 1993; Racault et al., 2012; Winder and Cloern, 2010) but have also tried to explain their spatio-temporal variability. Mixed Layer Depth (MLD) is recognized to be one of the main factors influencing phytoplankton phenology as it controls both light and nutrient availability for phytoplankton growth (Mann and Lazier, 2006). Numerous studies have attempted then to explain variability in phytoplankton phenology with MLD (Lavigne et al., 2013; Henson et al., 2006; Zhai et al., 2011) or with factors influencing it (Follows and Dutkiewicz, 2001, Ji et al., 2008). These studies have shown that, generally, a deeper winter mixing results in a delayed and stronger bloom in subtropical regions but in a weaker bloom in subpolar regions. Nevertheless, in most cases, MLD has failed to explain the overall phytoplankton variability suggesting that all the factors that impact phytoplankton phenology are still not known.

In oligotrophic environments, where nutrients are the primary factor controlling phytoplankton growth, changes in the subsurface distribution of nutrients should also affect phytoplankton phenology. Nutrient distribution is controlled by the interplay between biogeochemical and physical processes and external sources (Williams and Follows, 2003). Among the physical processes, oceanic circulation, which at global scale contributes to the transport of nutrients, can also regionally uplift or depress the nutricline and modify nutrient concentrations through cyclonic and anticyclonic circulation patterns (Williams and Follows, 2003). At basin scale, the trophic status of subtropical (subpolar) gyres is determined by anticyclonic (cyclonic) circulation, which controls the depth of the nutricline. At smaller scales, observations and modelling experiments have demonstrated that the formation of cyclonic eddies cause the upwelling of the nutricline just above the base of the productive layer resulting in an increase of the primary production in subsurface waters (McGillicuddy et al., 1998; Falkowski et al., 1991; Salihoglu et al., 1990). These observations were confirmed by a statistical analysis carried out in the Costa Rica Dome area (Kahru et al., 2007). In fact, the authors revealed a strong coupling between the Sea Level Anomaly (SLA) and the chlorophyll anomaly with positive chlorophyll anomalies associated with negative SLA. Regarding phytoplankton phenology, little is known about the impact of the cyclonic/anticyclonic circulation on it. To our knowledge, only two recent analyses have dealt with this point. A modelling analysis demonstrated that, in long lifetime cyclonic eddies of the Kuroshio current (lifetime superior than one year), the spring phytoplankton bloom is reinforced thanks to larger nutrient supply associated to vertical mixing (Sasai et al., 2010). In the North Western Mediterranean Sea, D'Ortenzio et al. (2014), using Bio-Argo data, suggested that large scale cyclonic circulation could be the first forcing factor of the

phytoplankton bloom, and demonstrated that deep winter convection is not essential for bloom development. To further investigate the influence of upwelling (downwelling) of the nutricline on the phytoplankton seasonal cycle, we present an analysis of the phytoplankton phenology during the cyclonic and anticyclonic phases of the North Ionian Gyre (NIG, see Figure 1). Thus, our aim is to determine whether and how changes in cyclonic/anticyclonic circulation can affect interannual variability in phytoplankton phenology.

The North Ionian Sea has been selected as study area because this region of the Eastern Mediterranean Sea has the specificity to experience reversals of its surface circulation at decadal time scale (Gacic et al., 2010). More specifically, due to the feedback with the Adriatic Sea (see below), the North Ionian circulation oscillates between the cyclonic and the anticyclonic modes (Figure 1b and 1c). Variations of horizontal circulation patterns have been detected mainly from altimetric measurements, while the vertical structure of the Ionian circulation has been inferred only recently using float and oceanographic campaign data (Gacic et al., 2014). It has been shown that the Ionian behaves as a two-layer fluid and that the decadal inversions of the NIG are generated by the spreading of the dense water formed in the Adriatic (AdDW). During the Eastern Mediterranean Transient, in the early 1990's, large amount of dense water of Aegean origin (about 3 Sv) contributed both to reinforce the anticyclonic circulation and to generate the reversal (Borzelli et al., 2009). Dense water of interannual varying density, changes the horizontal pressure gradient in the lower layer affecting the horizontal circulation over the entire water column. The mechanism at the base of the decadal reversal of the circulation regime was called Adriatic-Ionian Bimodal Oscillating System (BiOS) (Gacic et al., 2010). From a limited number of nitrate vertical profiles, Civitarese et al. (2010) showed that NIG circulation reversals influence North Ionian biogeochemistry and especially the depth of the nitracline. However, the impact on phytoplankton remained unclear. Comparing surface chlorophyll-a measured by the CZCS (Coastal Zone Color Scanner; 1979-1985) and SeaWiFS (Sea-viewing Wide Field-of-view Sensor, 1997-2000) ocean color sensors, D'Ortenzio et al. (2003) observed just a slight difference in the timing of main bloom events and, during the SeaWiFS period, a recurrent and large patch of [Chl-a]. More recently, Mayot et al. (2016) studied interannual variability in phytoplankton phenology in the Mediterranean Sea. Authors showed that phytoplankton phenology is extremely dynamic in the North Ionian Sea with significant changes in the annual cycle of surface chlorophyll-a concentration from year to year. However, they did not investigate the potential causes of this interannual variability. The present analysis attempts then to reconsider the impact of NIG circulation on phytoplankton phenology focusing on the satellite chlorophyll-a concentration (hereafter [Chl-a]) phenology and analyzing the 1998-2012 period.

In section 2, all the datasets used and the related methods are presented. Subsequently, temporal variability of the NIG circulation is analyzed and then the uplift (depression) of isopycnals and nitracline during the cyclonic (anticyclonic) regime is discussed. From the analysis of phenological metrics, differences are sought between cyclonic and anticyclonic regimes in [Chl-a] phenology. Phenological metrics are used here because they provide a synthetic method to highlight specific features of the [Chl-a] annual time series. These metrics also demonstrated to be extremely informative over different and various environments and regions (Follows and Dutkiewicz, 2001; Henson et al., 2009; Lavigne et al., 2013) despite the fact that they can be affected by data gaps (Cole et al., 2012) and noise (Ji et al., 2010). Focusing then on the start dates of [Chl-a]

increase, the impact of interannual variability in the North Ionian circulation and winter buoyancy fluxes are compared and discussed. Finally, hypotheses about causes and consequences of the observed patterns in [Chl-a] phenology are proposed.

## 2 Data and Methods

### 2.1 Satellite and modelling data

Daily [Chl-a] products, with a 4-km resolution, delivered by the Ocean Colour Climate Change Initiative (OC_CCI) project version 1.0 (http://www.esa-oceancolour-cci.org/) were used. The OC_CCI dataset, which covers the period September 1997 to July 2012, was created by band-shifting and bias-correcting MERIS and MODIS data to match SeaWiFS data and by merging the datasets with a simple average (Storm et al., 2013). Daily data were merged to obtain 8-day resolution maps. During the period from 1997 to March 2002 (only the SeaWiFS sensor available), 8-day average map time series contains

only 80% of good data but from April 2002, more than 96% of good data was available. To fill the gaps, a linear interpolation was applied to each time series. Then, the time series were smoothed with a 3-point median filter. OC_CCI [Chl-a] products are processed with the algorithm OC4v6 which has been fitted on a global dataset (O'Reilly et al., 2000). It has been demonstrated that this algorithm overestimates by a factor close to 2 [Chl-a] in oligotrophic conditions ([Chl-a] < 0.4 mg.m$^{-3}$) and alternative algorithms have been developed for the Mediterranean Sea (Bricaud et al., 2002; Volpe et al.,

2007). However, as well as the OC4 global algorithm, Mediterranean algorithms are based on the blue-green reflectance band-ratio model: [Chl-a] increases with a decrease of the blue-green band-ratio. Hence, similar patterns in [Chl-a] variations are obtained whith each class of algorithms. As the present study focus only on the intercomparison of [Chl-a] patterns in the Ionian Sea, the utilisation of global algorithm is valid. In addition, it allows the comparison with reference papers in Mediterranean phytoplankton phenology (D'Ortenzio et al., 2009; Mayot et al., 2016) which are also based on global [Chl-a]

products.

The Ssalto-Duacs Delayed-Time maps of the Absolute Dynamic Topography (ADT) (Rio et al., 2007) for the Mediterranean Sea (1/8°) computed and distributed by AVISO (http://www.aviso.altimetry.fr/) were downloaded for the period 1993-2014. As proposed by Gacic et al. (2014), the sea level gradient between the north and the center of the NIG was used to determine the cyclonic/anticyclonic nature of the NIG circulation. In practice, the ADT difference between rectangles S2 (18.75°E-

19.25°E and 39.5-40°N, see Figure 1b) and S1 (18.5°E-19°E and 37.5-38°W, see Figure 1b) was computed. Then, the daily time series was smoothed with a 30-day moving average filter (dotted line Figure 2a). As the anticyclonic (cyclonic) circulation regime is characterized by a doming (depression) of sea level, it is characterized by positive (negative) ADT difference value.

Net air-sea heat fluxes of long-wave radiation and short-wave radiation, sensible heat flux, latent heat flux and precipitations

were obtained from the European Center for Medium Range Weather Forecast (ECMFW) ERA-Interim reanalysis (Dee et al., 2011). Data was available at 6-hour intervals and downloaded at 0.5°-resolution. Then, daily averages were computed and the net heat flux was calculated as the sum of the four air-sea flux components. The surface buoyancy flux per unit area

was computed according to Anderson et al. (1996), with coefficients proposed by Gacic et al. (2002) for the Mediterranean Sea. Total winter buoyancy losses were then calculated as the time-integral for the period from 21 December to 20 March.

Daily temperature and salinity profiles for the product MEDSEA_REANALYSIS_PHY_006_004 produced by the Mediterranean Forecasting System and delivered by the CMEMS service (http://marine.copernicus.eu) were used to derive the depth of the isopycnal 28.9 kg m$^{-3}$ and MLD for the time range 1993-2012. MEDSEA_REANALYSIS_PHY_006_004 product results from a hydrodynamic model, supplied by the Nucleus for European Modelling of the Ocean (NEMO), with a variational data assimilation scheme for temperature and salinity vertical profiles and satellite Sea Level Anomaly along track data. The model horizontal grid resolution is 1/16° and the unevenly spaced vertical levels are 72. An evaluation of the product performance showed that temperature and salinity present an RMS of 0.34° and 0.1 psu, respectively, on average on the water column (Fratianni et al., 2016).

## 2.2 In-situ data

71243 temperature and salinity profiles (52% from Argo floats) were collected from the Coriolis web portal (http://www.coriolis.eu.org/) during the period 1990 from 2012, in the region comprised between 36°N-40°N and 16°E-21°E. Only the best quality (Coriolis sensu) data were retained. Then, potential density profiles were calculated and the depth of the isopycnal 28.9 kg m$^{-3}$ was determined.

Nitrate concentration, temperature and salinity profiles from oceanographic cruises in the area of interest (36°N-40°N, 17°E-22°E) were obtained: from POEM-BC-O91 (October 1991, 55 profiles), POEM-BC-A62 (April-May 1992, 40 profiles) and EMTEC (April-May 1999, 20 profiles). From nitrate concentration profiles, the nitracline depth was estimated as the depth of the isoline 1µM (Cermeno et al., 2008, Lavigne et al., 2013; Pasqueron de Fommervault et al., 2015) and compared to the depth of isopycnals 28.9 kg m$^{-3}$ (Figure 3c). The isopycnal 28.9 kg m$^{-3}$ was chosen here because it is located within the transition layer between the Atlantic Waters (AW, density ranging between 27.4 and 28.6 kg m$^{-3}$) which are generally nutrient-depleted and the Levantine Intermediate Waters (LIW, density ranging between 29.0 and 29.1 kg m$^{-3}$) where nutrients accumulate (Malanotte-Rizzoli et al., 1997; Ribera d'Alcalà et al., 2003). Although it was demonstrated that phytoplankton growth is co-limited by nitrate and phosphate in the Eastern Mediterranean Sea (Psarra et al., 2005; Thingstal et al., 2005), only nitrate dynamic is considered here as the both elements co-varies and control phytoplankton growth (Ribera d'Alcalà et al. 2003) and as the quality of phosphate data in database was not good enough for such analyses.

## 2.3 Phenological metrics

A set of phenological metrics has been computed from each annual [Chl-a] remotely sensed time series. Annual time series were defined from July of year n to June of year n+1, in order to better display the winter-to-spring transition. For each pixel, 14 annual time series could be defined ranging from July 1998 to June 2012. Hereafter, the annual time series are referred to as year n+1 (e.g. 2000 for the annual time series 1999-2000).

Phenological metrics were defined to describe the main characteristics of the phytoplankton annual cycle. They were selected to focus on the [Chl-a] content and on the date of the main growing period. Metrics CHL_Year, CHL_March and CHL_Max refer to the yearly average [Chl-a], the March average [Chl-a] and the [Chl-a] annual maximum, respectively. CHL_March has been defined here because previous studies demonstrated that in the Mediterranean Sea trophic situation is highly variable in March, spatially and interannually ranging from oligotrophy to bloom conditions (D'Ortenzio et al., 2009; D'Ortenzio et al., 2003; Mayot et al., 2016). The bloom initiation was defined as the date of the maximum growth rate. The date of the maximum growth rate represents an objective and efficient way to determine bloom onset date, although the methodology can be affected by data noise and requires then an efficient smoothing of the time series (Brody et al., 2013). The 8-day annual time series were twice filtered with running means using a 3-point window which decreases small scale noise without strongly degrading the main pattern of the [Chl-a] annual time series. Then, the middle date of 8-day interval corresponding to the maximum [Chl-a] increase was defined as Date_GR_Max. Finally, the metric indicating the date of maximum [Chl-a]is referred in the following by Date_Max.

## 3 Results and Discussion

### 3.1 Physical and chemical characterization of the NIG

The time series of ADT differencebetween regions S1and S2 (Figure 1b) shows decadal and seasonal variability (Figure 2). Globally, a first anticyclonic phase is evidenced from altimetry data (positive ADT difference) in 1992-1997 (Figure 2a). The anticyclonic phase is then followed by a cyclonic period (negative ADT difference) from 1998 through 2005. Then, a shorter anticyclonic period took place between 2006-2010. In 2012, the circulation was again cyclonic. To determine the average circulation state during each bloom, the ADT difference values were averaged for a period ranging from October to March which largely covers the annual [Chl-a] increase (see Figure 4). The resulting values (grey circles on Figure 2a) are referred to ICI (Ionian Circulation Index) in the following. Although, the ICI shows an interannual variability similar to the annual averages (July to June, black diamonds in Figure 2a), it is affected by seasonal signal which has to be considered for the determination of the circulation regime during transitional phases (i.e., when ADT differences are close to zero). During these phases, the Oct.-March ADT maps (Figure 2, panels b to f) were also investigated since the impact of the small scale structures on the ADT difference criteria could be substantial. From 2005 to 2006, the circulation turned from cyclonic to anticyclonic. In Figure 2b, the north of the Ionian domain is dominated by negative ADT values and, in the 34°-36°N band, a clear front, representing the Mid-Ionian Jet is observed. These patterns indicate a cyclonic circulation. In 2006 (see Figure 2c), high ADT values extend northward and the lowest ADT values are pushed along Italian and Greek coasts. This pattern represents the transition towards an anticyclonic regime, which is characterized by a higher ADT value in the center of the North Ionian basin than at its periphery. From Figure 2d to Figure 2f, the transition from an anticyclonic (Figure 2d) to a cyclonic (Figure 2f) circulation regime is represented. Indeed, similarly to Figure 2c, in Figure 2d a tongue with higher ADT values extents northward from the Mid Ionian Jet to the center of the NIG. Finally, the analysis of the ICI time series as well

as the examination of selected ADT brought us to the conclusion that from July 1998 to June 2005 and from July 2011 to June 2012, NIG circulation can be considered as cyclonic and that the periods 1993-1997 and July 2007-June 2010 are clearly dominated by the anticyclonic regime. No definite circulation regime can beattributed to the years 2006 and 2011 as they correspond to transitional phases.

In the Coriolis database, 26318 potential density profiles were available to describe the anticyclonic situation (data collected between 1990 and 1996 and between 2006 and 2009) and 34753 to describe the cyclonic situation (data collected between 1998 and 2005 and between 2011 and 2012). Optimal interpolation applied to each dataset provides an estimation of the depth of the isopycnal 28.9 kg m$^{-3}$ for cyclonic and anticyclonic regimes (Figure 3 panels a and b, respectively). The resulting maps exhibit very different features. In the cyclonic pattern (Figure 3a), the depth of the isopycnal 28.9 kg m$^{-3}$
ranges between 40 and 120m north of 37°N. South of 37°N, a steep gradient, implying isopycnal southward deepening, is observed consistently with the position of Mid-Ionian Jet which is reinforced during the cyclonic regime of the NIG (Bessieres et al., 2013; Gacic et al:, 2014). The shallowest isopynals are located around 39°N and 18°E, offshore of the Italian coast, representing the center of the cyclone. During the anticyclonic phase, the isopynal 28.9 kg m$^{-3}$ is very deep (240m) in the center of the Ionian Sea and is shallower at the border of the basin. This isopycnal depth pattern is consistent
with an anticyclonic circulation.

As previously demonstrated for the global ocean (While and Haine 2010, Omand and Mahadevan, 2013), the depth of an isopycnal can be representative of the nitracline depth in case of stratified water column. This is confirmed in the North Ionian from the obtained linear relationship between the depth of the nitracline and the 28.9 kg m$^{-3}$ isopycnal (Figure 3c, r$^2$=0.93). Data used here were collected at different seasons (i.e. in October 1991 and April/May 1992 and 1999) but always
during the long period during which water column is stratified. As soon as vertical mixing reaches nitracline depth, this relationship is not anymore valid. Nevertheless, such events are rare and short in the North Ionian Sea. Applying this relationship, it appears that in the center of the North Ionian, the nitracline depth ranges between 60 and 80m during the cyclonic regime whereas it reaches 150m when the NIG is anticyclonic. Considering that this nitracline depth range (i.e. 60 to 150 m) overlaps with climatological values of winter MLDs in the Ionian Sea (i.e. 80-130m; D'Ortenzio et al., 2005;
Houpert et al., 2015), changes in NIG circulation should have significant impact on the entrainment of nutrientsto the surface layer by mixing and, consequently, on phytoplankton growth and primary production.

### 3.2 General patterns of phytoplankton phenology in the Ionian Sea

Phytoplankton phenology in the Ionian Sea is typical of very oligotrophic regions with low [Chl-a] (values range between 0.05 and 0.5 mg m$^{-3}$in open waters) and a seasonal cycle characterized by very low [Chl-a] values during summer and higher
values during winter (Figure 4; D'Ortenzio and Ribera d'Alcalà, 2009). The highest [Chl-a] values are observed along the coast, especially along the South Italian coast where a local upwelling can take place (D'Ortenzio et al. 2003). Offshore, a north-south gradient is observed (Figure 4 and 5a) with higher values in the northern part of the basin (above 38°N). In the whole basin, the lowest [Chl-a] are observed in mid-summer, then, [Chl-a] increases starting in October (Figure 4).

Date_GR_Max occurs later, in December for the central Ionian Sea and in mid-January for the North Ionian and for the Italian coastal region (see Figure 5). Spatial variability in the Date_GR_Max indicates that differences in [Chl-a] phenology should exist between the northern and southern parts of the basin. This is confirmed by the analysis of the seasonal [Chl-a] pattern in the Hovmöller diagram (Figure 4). Indeed, during some winters as for example 1999, 2000, 2003, 2005, there is a

distinct local maximum in [Chl-a] generally centered in March. These March peaks are situated in the 38°N-39°N band and are never observed below 37°N. However, they are not observed in all situations and, for instance, no [Chl-a] peak appears in March 2001, 2007 or 2008. This special feature of the [Chl-a] seasonal cycle is related to the location of the isopycnal doming area during cyclonic regime (Figure 3a), suggesting that interannual variability in [Chl-a] patterns in this region may be related to NIG circulation pattern.

**3.3 Impact of the NIG circulation on the [Chl-a] phenology**

Figure 5 displays the July 1998 – June 2012 average of phenological metrics (left panels) as well as the anomalies for the cyclonic (1999-2005 and 2012), and the anticyclonic (2007-2010) regimes compared to such interannual average. Regarding [Chl-a] content, no significant difference is observed for the annually averaged [Chl-a] content (i.e. CHL_Year), with indeed a relative difference ranging between -10% and +10%. During the anticyclonic regime, a small positive anomaly is generally

observed along the coast while central Ionian displays negative anomalies. When March only is considered, the CHL_March displays much stronger differences between cyclonic and anticyclonic regimes (panels e and f). Significant differences are observed in the center of the North Ionian Sea with a +15% anomaly during the cyclonic regime and a -20% anomaly during the anticyclonic regime. In the South Ionian, no major differences are observed in March, which is in agreement with the low interannual variability observed below 37°N (Figure 4). Similarly to CHL_March, CHL_Max shows significant differences

in the central North Ionian with a positive or a negative anomaly during cyclonic and anticyclonic regimes, respectively. In South Ionian, the same feature is observed due to variability of the AW advection modulated by the BiOS. Regarding metrics measuring the timing of the bloom, significant differences are again observed in the center of the North Ionian. Indeed, Date_Max and Date_GR_Max are delayed by about 20 and 30 days, respectively, when NIG circulation is cyclonic. On the other hand, both are early by more than 40 days when the circulation is anticyclonic.

The present analysis suggests that in the North Ionian, two types of [Chl-a] phenology co-exist, one characterizing the cyclonic regime and the other one characterizing the anticyclonic regime. This is also confirmed by the analysis of annual [Chl-a] cycles in the region S3 (17.5°E-18.5°E, 38°N-39°N, Figure 6). We have chosen the position of the area S3 where to calculate spatially averaged phenological metrics so that it coincides with the region where the extremes of [Chl-a] anomalies occur (Figure 5). During the anticyclonic regime (2007-2010), all cycles show a regular increase in [Chl-a] from

October to the end of January. Maximum growth rate appears between late November and late December, and maximum [Chl-a] between mid-January and mid-February (Table 1). D uring the cyclonic regime, as for the anticyclonic one, a first increase in [Chl-a] occurs in mid-September and extends to January. In March, most of the time series display a narrow and strong peak in [Chl-a]. The annual maximum growth rate is clearly associated to this March peak occurring generally a few

days (10 days in average) before the date of the [Chl-a] maximum. Although, the cyclonic mode is characterized by the March [Chl-a] peak and a higher amplitude of the annual [Chl-a], this does not significantly impact the yearly average [Chl-a].

Referring to the climatological analysis of the shapes of [Chl-a] annual cycles proposed by D'Ortenzio and Ribera d'Alcalà (2009), the anticyclonic cycles (Figure 6) are close to the so-called "NO BLOOM" shape whereas most of the cyclonic cycles could be assimilated to the "INTERMITTENT" [Chl-a] annual cycle. Lavigne et al. (2013) interpreted the "INTERMITTENT" dynamics as an intermediate state between the "NO BLOOM" dynamics and the typical dynamics of spring bloom as observed in the north-western Mediterranean Sea (Marty et al., 2002). The "NO BLOOM" shape was identified in areas where winter mixing rarely reaches the nitracline depth and where light is always sufficient to support phytoplankton growth (Lavigne et al., 2013).

## 3.4 Role of the NIG circulation patterns compared to the interannual variability in MLD (focus on the region S3)

In the North Ionian where the bimodal behavior of the circulation (i.e. cyclonic versus anticyclonic) has a direct influence on phytoplankton phenology (see Section 3.3), interannual variability also exists independently of the BiOS switch (see Table 1 and Figure 6). Still, focusing on the area S3, the interannual variability of phenological metrics is compared to variability of the ICI (see definition Section 3.1) and total winter buoyancy losses. The total winter buoyancy losses are considered here because they largely contribute to the formation of the mixed layer (Gill, 1982) and are thus expected to affect bloom development (Follows and Dutkiewicz, 2001). Regarding the cyclonic regime, the main feature in [Chl-a] phenology is the absence of a late [Chl-a] peak in March in 2001 and 2002 (Figure 6, Table1). Years 2001 and 2002 exhibit the lowest values for CHL_March and CHL_Max and more importantly, the earliest dates for Date_Max and Date_GR_Max which occurred as earlier as January-February and December-January, respectively (Table 1). This special pattern may be explained by the buoyancy losses which were weak:-24% (2001) and -35% (2002) compared to the reference 1999-2012 average (see Table 1). In 2001 and 2002, a shallow winter MLD probably prevented significant nutrient inputs to the surface layer and consequently hindered the spring phytoplankton bloom (see discussion below). On the other hand, in 2005 and 2012 winter buoyancy losses were particularly high (+42%, +43%, respectively) and CHL_Max and CHL_March show the highest values observed. More generally, we observed a positive Spearman correlation between the buoyancy flux anomaly and CHL_March (correlation = 0.79) or CHL_Max (correlation = 0.89), suggesting that in addition to the circulation regime, winter buoyancy losses affect the presence and the strength of the [Chl-a]. Regarding the anticyclonic regime and the transitional years 2006, 2011, a very low interannual variability is observed: only the year 2006 (transitional regime) displays a [Chl-a] peak in early March (Table 1) which coincides with an ICI value close to 0 (transitional regime) and a weak positive anomaly in buoyancy losses (+6%, Table 1). During the anticyclonic regime, no significant correlation was found between buoyancy losses and phenological metrics. To sum up, in the center of the North Ionian, both the circulation patterns and air-sea exchanges would affect [Chl-a] cycles. Nevertheless, it seems that the impact of air-sea exchanges may be stronger when circulation is cyclonic and apparently affects the amplitude of the March peak.

Given the general oligotrophy of the Eastern Mediterranean (Ribera d'Alcalà et al., 2003), horizontal nutrient advection in the Ionian Sea should not have any significant impact on the late winter/early spring [Chl-a] peak observed during the cyclonic regime. However, the analysis of Table 1 as well as the spatial correspondence between phenological anomalies and isopycnals uplift/depression suggest that, in the North Ionian Sea, the March bloom would be favored by the cyclonic

circulation which tends to uplift isopynals and nitracline depth towards the surface and by strong negative winter buoyancy fluxes which contribute to deepen the MLD. In other words, North Ionian phytoplankton phenology is driven by the interactions between the nutrient field and MLD which is consistent with sub-tropical regions where phytoplankton growth is almost permanently limited by nutrient availability (Longhurst, 2006). From these results, we propose two scenarios to explain interannual changes in phytoplankton phenology in the North Ionian Sea. These scenarios are based on relative

position of the nitracline and the winter MLD. In the first scenario (scenario A), the deepest winter MLD remains above the nitracline depth and there is no spring phytoplankton bloom whereas, in the second scenario (scenario B), winter MLD lies deeper than the fall nitracline depth which drives remarkable nutrient supplies to the surface and supports the development of a March bloom. The first scenario results then in a deep nitracline depth and/or a shallow MLD. Conversely, scenario B occurs if the nitracline depth is shallow and/or if the MLD is deep.

Scenario B was observed in some subtropical regions, like the BATS station in the Subtropical North Atlantic (Steinberg et al., 2001). There, vertical mixing represents one of the major processes to supply nutrients to the illuminated surface layer (Steinberg et al., 2001; Hansell and Carlson2001). At lower latitude and generally in Trades' biome regions according to Longhurst (2006), trophic regimes close to scenario A are observed. For instance, at the station HOT in the North Pacific Subtropical gyre (Karl and Lucas, 1996), winter mixed layers rarely reach the deep nutrient repleted waters and a deep

chlorophyll maximum is almost permanent (Letelier et al., 2004). The rate of primary production is then explained by stochastic nutrient injections into the euphotic zone caused, for instance, by internal waves, cyclonic mesoscale eddies or wind-driven Ekman pumping (Hayward, 1991; Karl 1999). In the Mediterranean Sea, Lavigne et al. (2013) showed that in the South Ionian Sea and in the Levantine Sea, the probability that the MLD reaches the nitracline depth is very small. These results were corroborated by a recent analysis based on Bio-Argo floats (Pasqueron de Fommervault, 2015). However, in

regions where meso-scale circulation is cyclonic (e.g. South Adriatic gyre, Rhodes gyre), there are evidences of surface nutrient enrichment caused by vertical mixing (Gacic et al., 2002; Lascaratos et al., 1993; Salihoglu et al., 1990). Thus, the Eastern Mediterranean Sea likely holds both trophic regimes; one involving subsequent surface nutrient supplies by winter vertical mixing and the other not receiving such supply.

Figure 7 supports our previous hypothesis to explain interannual variability in phytoplankton phenology. Indeed, during the

anticyclonic phases of the circulation regime (i.e.1993-1997 and 2007-2010), estimated nitracline depth was relatively deep and ranged between 100m and 150m. The nitracline is never reached by the MLD estimated with the difference density criterion of 0.005 kg m$^{-3}$ (Brainerd and Gregg, 1995), and was rarely reached by the deepest MLD estimated with the difference density criterion of 0.03 kg m$^{-3}$ (de Boyer-Montégut et al., 2004). In addition, during the anticyclonic period, the isopycnal 28.9 kg m$^{-3}$ never intersected the surface, indicating that winter mixing is not able to penetrate this layer. During

the cyclonic phases (1999-2005 and 2012), the nitracline was shallower (generally less than 100m). Except for 2001, the shallowest MLD estimations exceeded the fall nitracline depth and reached almost the surface indicating a penetration of mixing down to the isopycnal 28.9 kg m$^{-3}$ layer and a possible nutrient enrichment.

A common explanation for the fall and spring bloom is the succession of a entrainment bloom (i.e. bloom driven by MLD deepening in nutrient-limited conditions) in fall followed by a spring bloom (light-limited, Cullen et al. 2002; Levy et al., 2005). The onset of the spring bloom, which is initially prevented by a deficiency of light in the surface mixed layer, is generally explained by a stratification or a stabilization of the water column. Stratification can be caused by the shoaling of the mixed layer (Sverdrup, 1953), or, as recently suggested, by submesoscale eddies (Mahadevan et al., 2012). Other studies have shown that a phytoplankton bloom can be induced by a decrease of the water column turbulence (Huisman et al., 1999) in response, for instance, to the inversion of the surface heat fluxes (Taylor and Ferrari, 2011). However, previous hypotheses about fall and spring phytoplankton blooms were developed for temperate and sub-polar regions and do not fit with the light and nutrient conditions of the Ionian Sea. Indeed, on one hand, in the Ionian Sea, winter solar irradiation is sufficient for supporting phytoplankton growth, according to the Sverdrup (1953) approach (Lavigne et al., 2013). On the other hand, Figure 7 shows that when the first Chl-a increase is observed in fall (October to December), the MLD is shallower than the nitracline depth making the entrainment bloom hypothesis an unconvincing explanation for the fall [Chl-a] rise. A possible explanation for the early increase in [Chl-a] could be sought in the seasonal increase of Chl-a content per cell in response to a diminution of the light intensity (Winn et al., 1995; Behrenfeld et al., 2005). Indeed, from a modelling approach, Taylor et al. (1997) showed that the phytoplankton carbon to chlorophyll ratio varies by a factor of 2 in subtropical regions (~35°) from early fall to December.

Spring peaks in [Chl-a], mainly observed during cyclonic circulation regime, can be considered as typical spring blooms (i. e. detrainment blooms, Cullen et al., 2002) which are light-driven. Indeed, Date_GR_Max was sometimes observed during the MLD shallowing. However, the tight coupling between MLD deepening (deeper than nitracline) and [Chl-a] increase as well as the lack of light limitation in the Eastern Mediterranean (Lavigne et al., 2013), induce us to conclude that [Chl-a] increase is the result of a late winter entrainment bloom (nutrient-driven). In addition, as the date of the bloom is determined from surface [Chl-a], the dilution effect during the mixed layer deepening masksthe phytoplankton accumulation, causing an artificial delay in the computed date of the bloom (Behrenfeld et al., 2010).

## 4 Summary and Conclusion

In this study we show that in the Ionian Sea two trophic regimes co-exist. In one case, the absence of spring bloom peak in March indicates the lack of nutrient supply to the surface layer. In that case, the rate of regenerated production is high year-round and new primary production would only be sustained by sporadic nutrient injections through the nutricline. In the second case, the presence of a [Chl-a] peak in March could be explained by a significant nutrient supply by vertical mixing which sustains the phytoplankton bloom and the seasonal boost in new primary production. We showed that these

differences in the phytoplankton dynamics were associated with the variability of the circulation regime of the NIG. Namely, the March [Chl-a] peak was commonly observed during the cyclonic phases, whereas no peak was observed in case of anticyclonic circulation. In addition to the strength and sense of the circulation, the [Chl-a] annual cycle is affected also by the winter buoyancy losses, which determine the MLD. Thus, it appears that phytoplankton phenology is the result of the interplay between surface circulation and atmospheric conditions, affecting the nitracline depth and the MLD. Hypotheses based on the relative position of the nitracline depth and the winter deepest MLD were then proposed to explain observed phenological patterns.

The dependence of phytoplankton phenology on the circulation regime in the Ionian (driven by BiOS) establishes a link between phytoplankton dynamics and the semi-closed thermohaline cell of the Eastern Mediterranean. As shown previously (Gacic et al., 2014), the Adriatic Sea, where the thermohaline cell of the Eastern Mediterranean originates, is highly sensitive to the winter air-sea heat fluxes: for example, the exceptionally severe winter of 2012 resulted in the occasional production of a large amount of very high dense water in the Adriatic, whose spreading in the deep Ionian caused a partial reversal of the circulation.

In conclusion, future studies on phytoplankton dynamics in the Ionian Sea should not take into account just the local circulation regime, but winter climatic conditions over the Adriatic Sea as well.

**Aknowledgements**

The authors thank the ESA Ocean Colour CCI Team for providing OC-CCI chlorophyll data; NASA for providing SeaWiFS, MODIS and MERIS chlorophyll data; ECMWF for providing meteorological data and AVISO with support from CNES for distributing altimeter products produced by Ssalto/Duacs. The Coriolis Centre for in situ Oceanographic Data and the National Oceanographic Data Center of the OGS (NODC-OGS) are acknowledged for the collection of the CTD and nitrate profiles that were made available to this study. This study has been conducted using E.U. Copernicus Marine Service Information. We are also grateful to DhyanAranha for proof-reading this manuscript. We acknowledge the European Commission "Cofunded by the European Union under FP7- People - Co-funding of Regional, National and International Programmes, GA n. 600407" and of the RITMARE Flagship Project, for the support to H. L. during her stay at OGS. This work is also the contribution to the French "Equipementd'Avenir" NAOS project (ANR J11R107-F) and to the PERSEUS EU Project. M.G. was partially supported by Croatian Science Foundation under theprojectSCOOL (IP-2014-09-5747). Two anonymous reviewers are thanks for their constructive comments on a first draft of the manuscript.

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

**Table 1: Average Ionian Circulation Index (ICI, ADT difference values between regions S1 and S2 averaged overOctober-March), winter buoyancy fluxes (WBF) and phenological metrics in the region S3. ICI is expressed in cm and WBF are expressed in percent relative difference compared to the 1999-2012 interannual average. CHL_Year (yearly average [Chl-a]), CHL_March (March average [Chl-a]) and CHL_Max (annual maximum [Chl-a]) are expressed in mg m$^{-3}$. Numbers in brackets are standard deviations. For Date_Max (date of annual maximum [Chl-a]), Date_GR_Max(date of maximum growth rate in [Chl-a]) standard deviations are expressed in days.**

| year | regime | ICI | WBF | CHL_Year | CHL_March | CHL_Max | Date_Max | Date_GR_Max |
|------|--------|-----|-----|----------|-----------|---------|----------|-------------|
| 1999 | Cyclonic | -3.522 | -7.7 (2.9) | 0.19 (0.011) | 0.36 (0.07) | 0.45 (0.12) | 16-March (37) | 04-March (39) |
| 2000 | Cyclonic | -4.186 | 26.2 (3.2) | 0.18 (0.008) | 0.30 (0.04) | 0.46 (0.16) | 02-April (29) | 12-March (38) |
| 2001 | Cyclonic | -1.826 | -23.8 (0.8) | 0.15 (0.008) | 0.17 (0.01) | 0.29 (0.03) | 19-Jan. (34) | 30-Dec. (28) |
| 2002 | Cyclonic | -2.905 | -34.9 (0.6) | 0.17 (0.011) | 0.26 (0.03) | 0.33 (0.05) | 28-Feb. (14) | 31-Jan. (41) |
| 2003 | Cyclonic | -3.966 | 5.3 (0.6) | 0.17 (0.008) | 0.26 (0.03) | 0.39 (0.06) | 02-April (2) | 20-March (15) |
| 2004 | Cyclonic | -2.395 | 15.2 (2.2) | 0.17 (0.008) | 0.34(0.04) | 0.46 (0.10) | 24-March (6) | 20-March (6) |
| 2005 | Cyclonic | -1.281 | 42.2 (3.1) | 0.18 (0.013) | 0.44 (0.06) | 0.90 (0.19) | 24-March (2) | 20-March (3) |
| 2006 | Transition | +2.49 | 6.2 (1.8) | 0.17 (0.014) | 0.28 (0.03) | 0.34 (0.06) | 08-March (22) | 31-Jan. (31) |
| 2007 | Anticyclonic | +7.5 | -17.3 (2.0) | 0.14 (0.019) | 0.20 (0.03) | 0.24 (0.04) | 20-Feb. (26) | 20-Nov. (39) |
| 2008 | Anticyclonic | +5.53 | -23.9 (2.5) | 0.16 (0.024) | 0.20 (0.08) | 0.32 (0.09) | 19-Jan. (28) | 30-Dec. (47) |
| 2009 | Anticyclonic | +2.73 | 12.4 (2.9) | 0.16 (0.019) | 0.20 (0.02) | 0.36 (0.08) | 04-Feb. (11) | 06-Dec.(18) |
| 2010 | Anticyclonic | +0.75 | -18.1 (3.2) | 0.19 (0.025) | 0.25 (0.03) | 0.36 (0.08) | 11-Jan. (11) | 14-Dec. (26) |
| 2011 | Transition | +0.28 | -26.3 (2.9) | 0.18 (0.015) | 0.22 (0.02) | 0.33 (0.07) | 18-Dec. (17) | 12-Nov. (22) |
| 2012 | Cyclonic | -3.795 | 44.6 (3.8) | 0.16 (0.010) | 0.35 (0.07) | 0.46 (0.13) | 24-March (6) | 03-Dec. (2) |

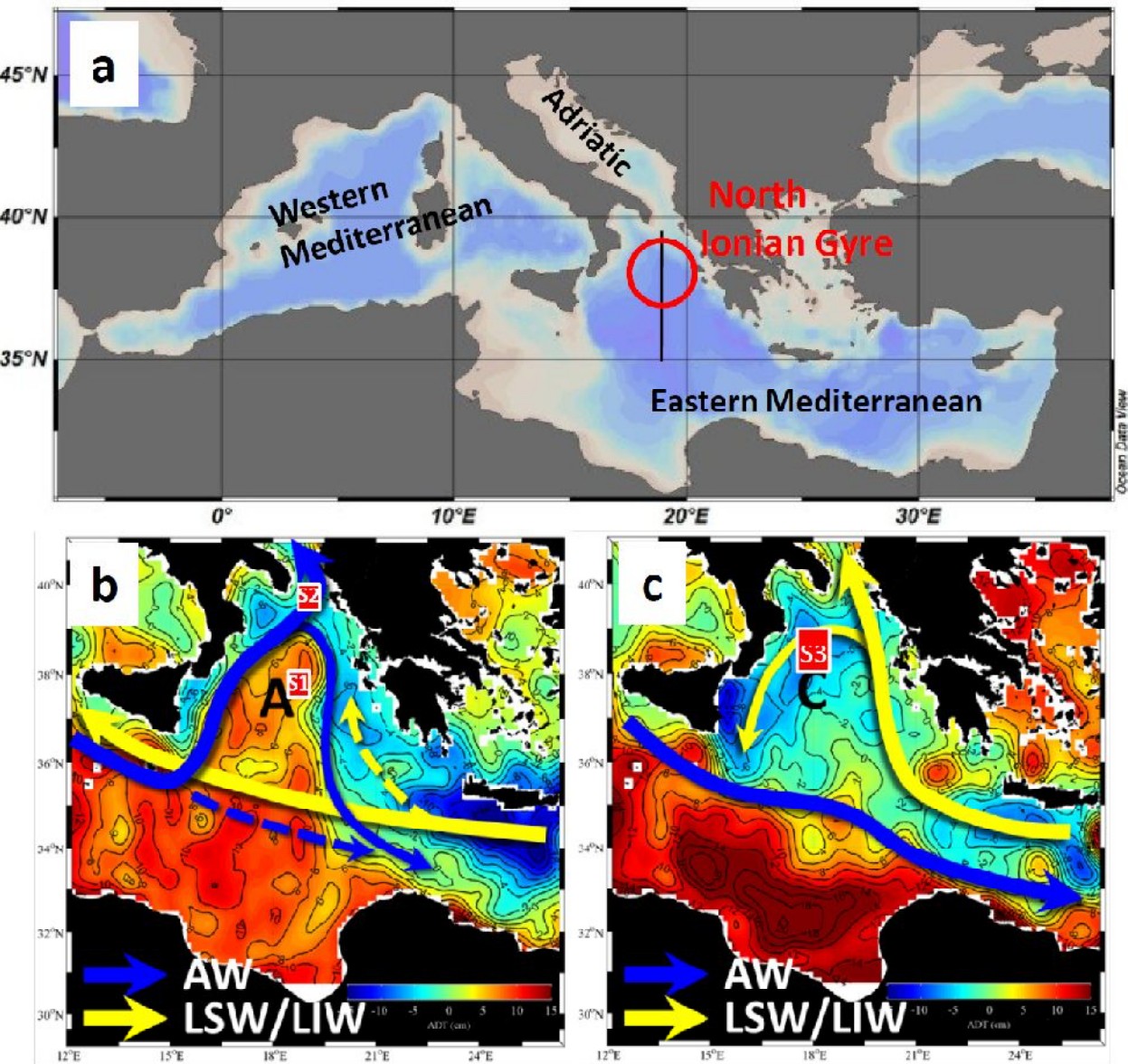

**Figure 1:** Study area (panel a) and schematic representation of the North Ionian Gyre upper-layer circulation during anticyclonic (panel b) and cyclonic (panel c) regimes. AW, LSW and LIW refer as Atlantic Water, Levantine Surface Water and Levantine Intermediate Water, respectively. See text for a definition of regions S1, S2 (Section 2.1) and S3 (Section 3.2). The underlying maps represent the 1995 (b) and 1999 (c) average absolute dynamic topography, respectively.

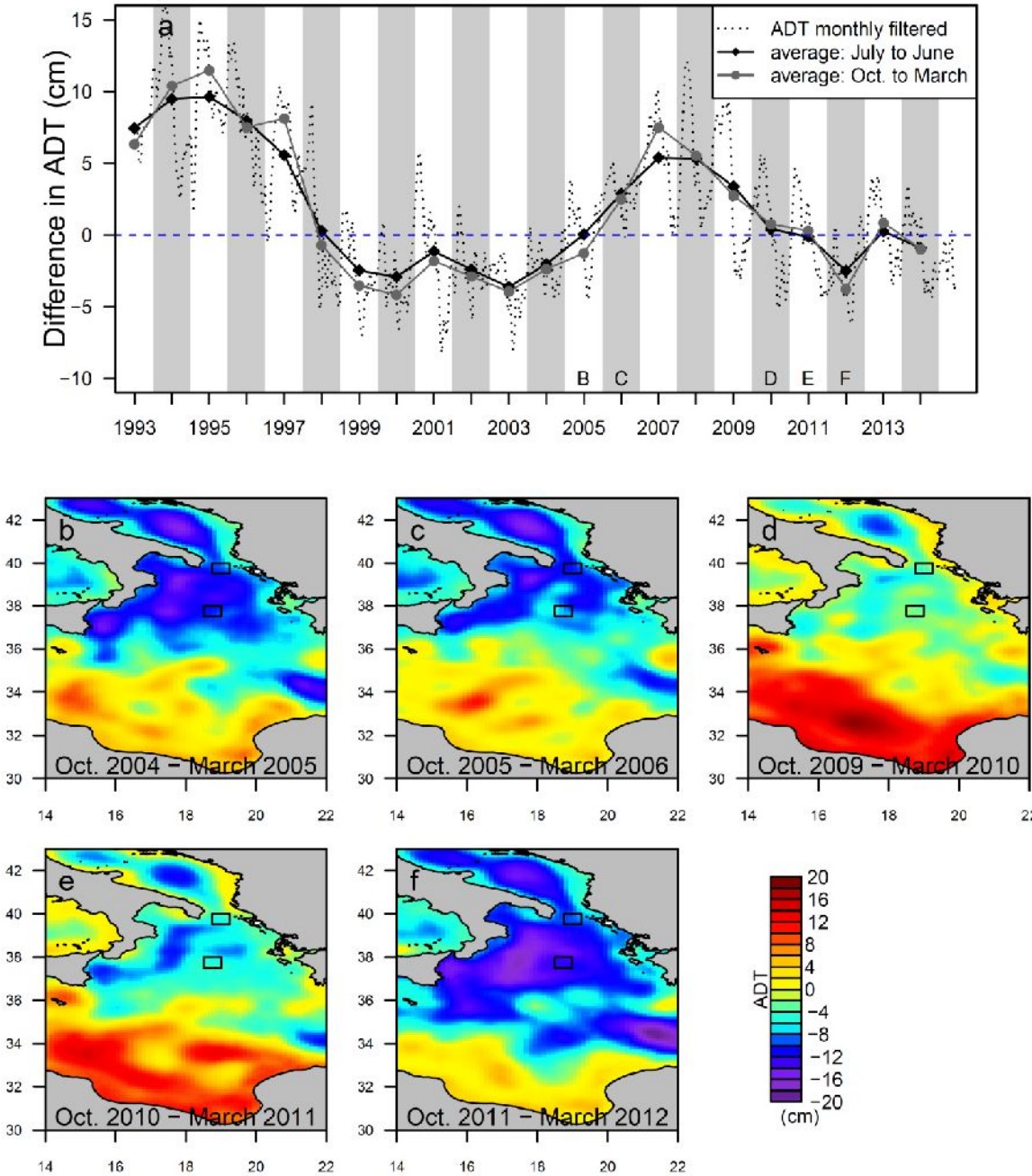

**Figure 2: Panel a: Dotted line is the ADT difference between regions S1 and S2 (S1 – S2, see Figure 1b). Annual periods used for the phenology analysis, which range from July of year n to June of year n+1 are featured by white and grey bands. For each of them, black diamonds and grey circles represent the averaged ADT difference during periods October to March (also named ICI) and July to June, respectively. The zero level differentiates cyclonic (negative values) and anticyclonic (positive values) regimes. Panels b to f represent the average ADT pattern for different periods. Squares represent the regions S1 (south) and S2 (north)**

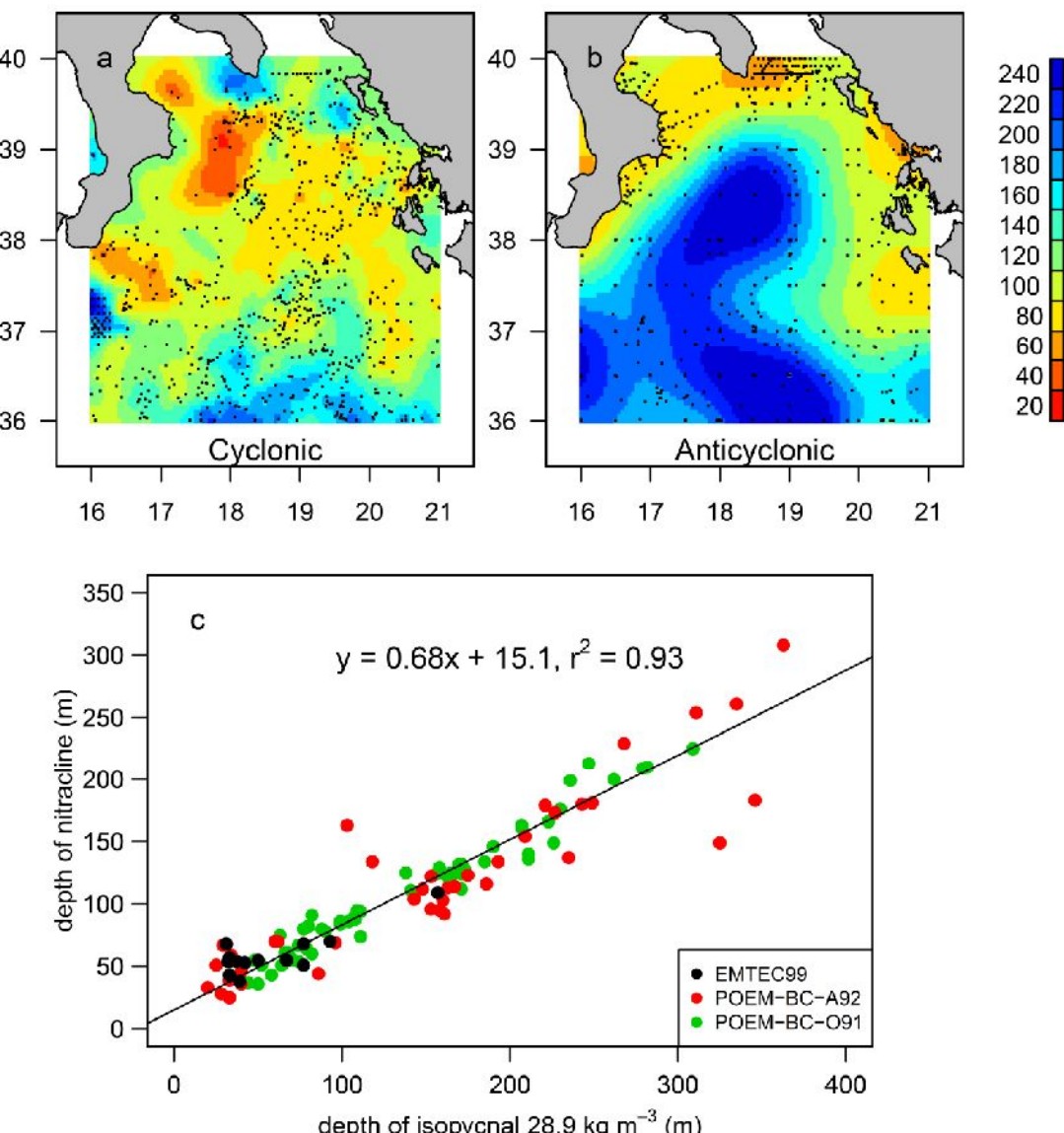

**Figure 3: Climatological maps of the estimation of the depth of the isopycnal 28.9 kg m$^{-3}$ for cyclonic (panel a) and anticyclonic (panel b) regimes. Data were spatially interpolated with kriging, black dots indicate in situ data sued for kriging. Relationship between the depth of the nitracline and the depth of isopynal 28.9 kg m$^{-3}$ in the North Ionian Sea (panel c) from in situ data collected during three oceanographic cruises (see colors).**

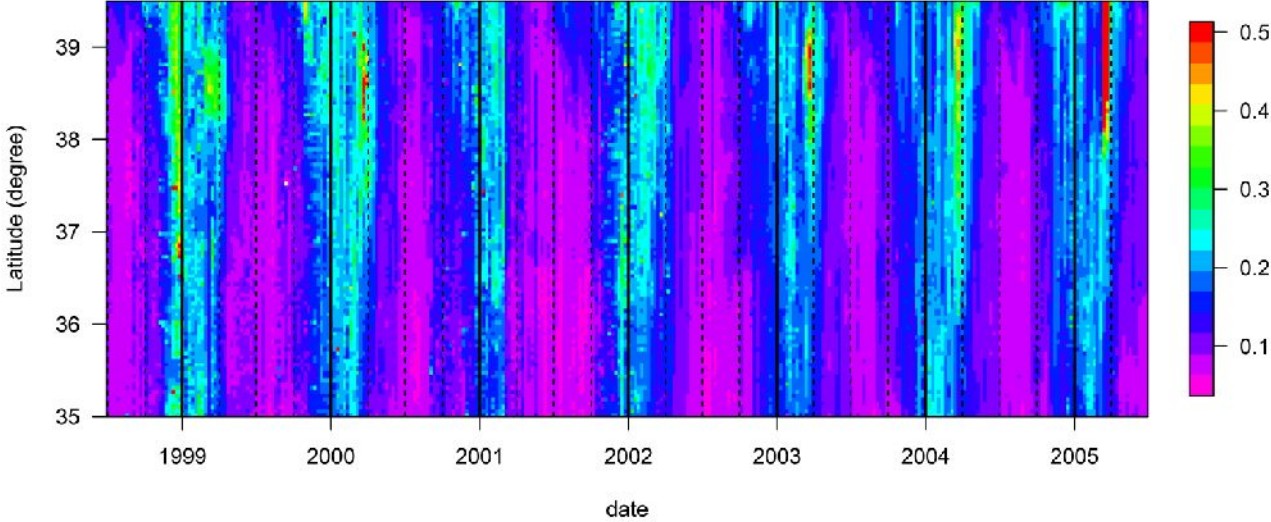

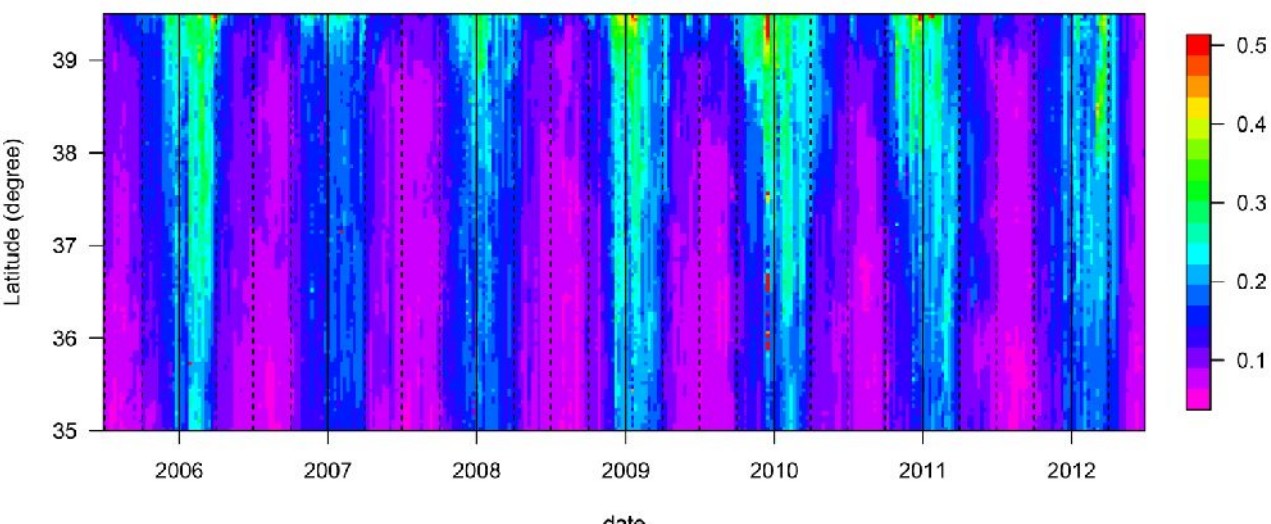

**Figure 4: Hovmöller diagram of the 8-day satellite [Chl-a] along the 18.5°E meridian (see vertical line in Figure 1a). [Chl-a] is expressed in mg m⁻³. Solid lines indicate 1-year periods (January to December) and dotted lines 3-month periods.**

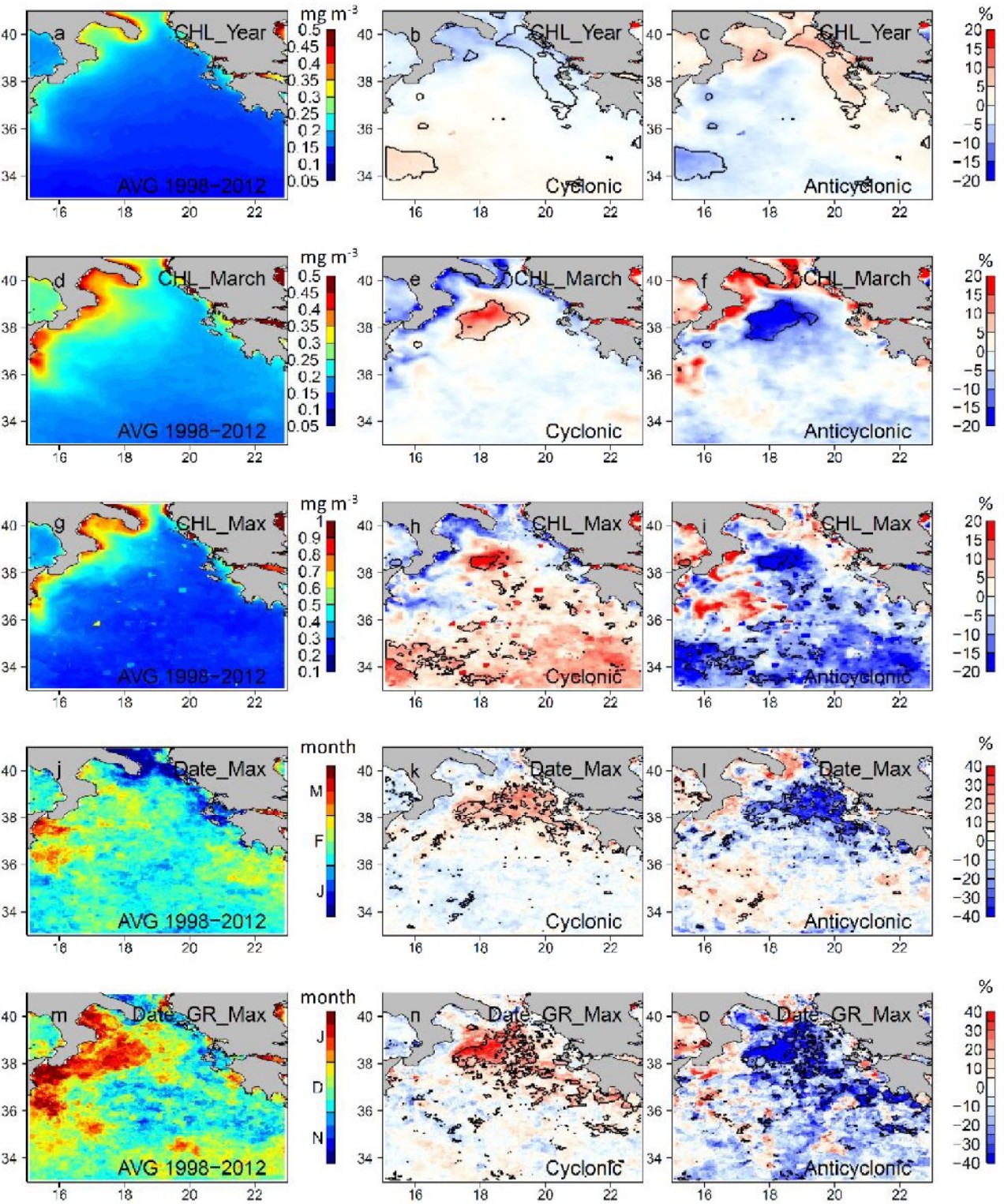

**Figure 5: left panels: interannual average values (1998-2012) of phonological metrics. CHL_Year, CHL_March and CHL_Max are expressed in mg m$^{-3}$. Middle and right panels show the anomaly of metrics compared to the interannual average for the cyclonic (middle panels) and anticyclonic (right panels) periods. They are expressed in percent difference for CHL_Year, CHL_March and CHL_Max, and in days for Date_Max and Date_GR_Max. Solid black lines refer to areas with significant difference between cyclonic and anticyclonic patterns (p-value < 0.05).**

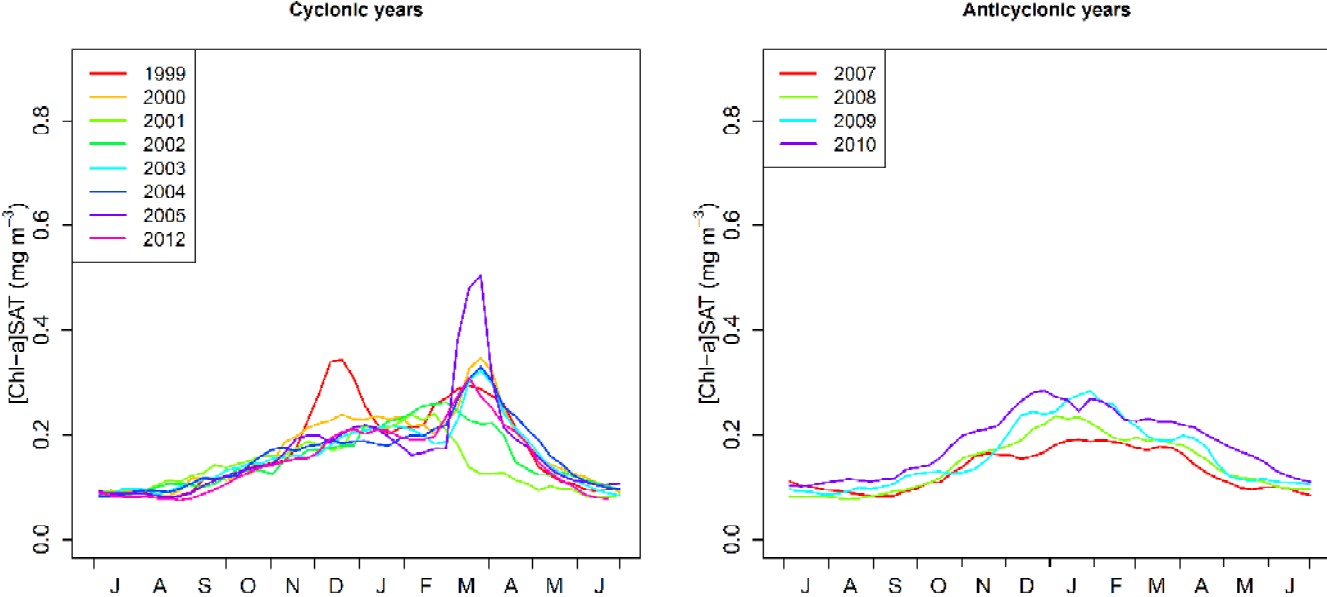

**Figure 6: [Chl-a] (mg m$^{-3}$) annual cycles for the cyclonic and anticyclonic regimesaveraged in the region S3 (see Figure 1).**

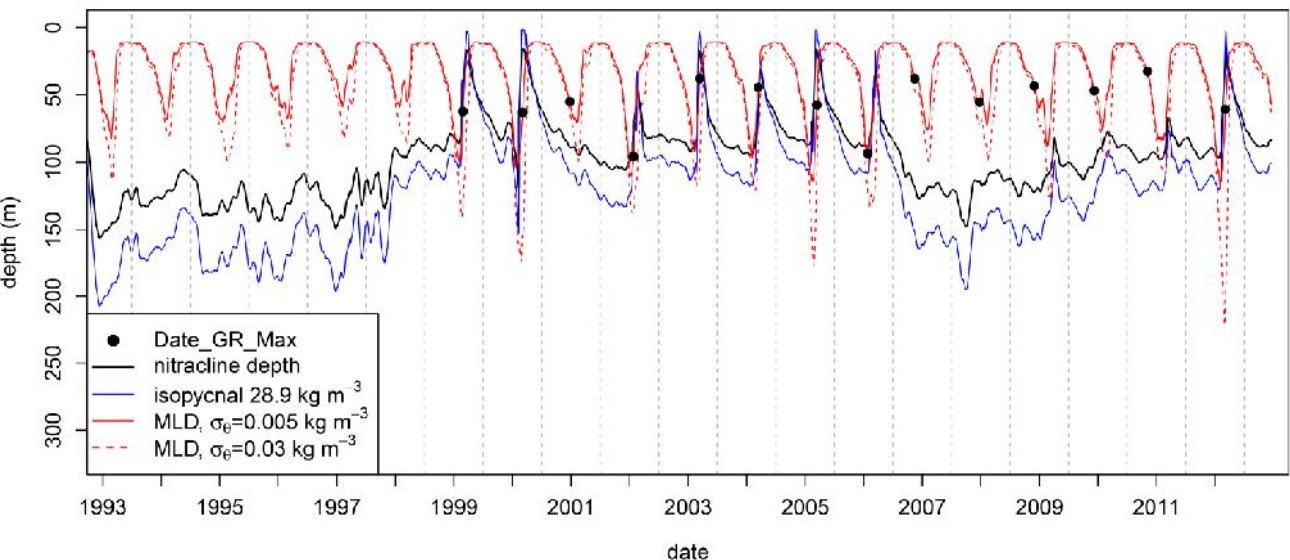

**Figure 7: Time series**, for the region S3, of the mean nitracline depth (solid black line), depth of the isopycnal 28.9 kg m$^{-3}$ (solid grey line) and MLD computed with a density criteria of 0.005 kg m$^{-3}$ (solid red line) and 0.03 kg m$^{-3}$ (dotted red line) from a surface reference of 10m. Nitracline depth was derived from the depth of the isopycnal 28.9 kg m$^{-3}$ with the linear relationship established on Figure 3c. Circles represent the mean Date_GR_Max dates for the region S3.