# Peer review of "Impact of decadal reversals of the North Ionian circulation on phytoplankton phenology"

_Biogeosciences, 2017_

## Referee Comment (RC1) · Anonymous Referee #1 · 22 Feb 2018

General comments

Several phytoplankton phenology parameters, such as the chlorophyll maximum value in March, chlorophyll max increase, etc. are obtained from satellite imagery of the Ionian Sea and related to both the documented effects of BiOS cyclonic-anticyclonic circulation reversals in the basin and air-sea fluxes. This is done to demonstrate the high impact of this physical forcing on the Ionian Sea productivity. I find this paper extremely interesting and ground-breaking. A just biological "add-on" to the knowledge concerning BiOS phenomenology. The Authors have resorted to a varied and complex dataset to illustrate their results, indicating also good interdisciplinary teamwork. The

text is well-written (needs English improvement, though) and structured. It is clear and results seem to me more than solid, and very interesting. Especially, it is a sort of "eye-opener" indicating that biological/trophic monitoring CANNOT be confined to the coasts, as often happens: basin-wide variability is just as important to gain insight on e.g. changes occurring in the marine trophic chain. A similar study on secondary productivity (or fisheries) would be extremely interesting. Therefore, I recommend publication, after the minor corrections reported below. The list is long but they are mainly linguistic corrections.

Form

The English of the manuscript is reasonable but needs improvement. I tried to help with the list of corrections suggested below.

Particular comments and suggested text corrections

Abstract

Line 12. Replace "regime with (hereafter "replace" = "->") "regimes". Lines 13-14. "the vertical dynamics and the nutrient distribution" -> "both vertical dynamics and nutrient distribution". Line 17. "time-series" -> "time series". No need for hyphen. Please change throughout the text. Line 19 and 20. "bloom initiation" -> "bloom onset". "Initiation" is more in a religious sense, in English. Line 20. "In the center of the gyre". Which gyre? Is it the North Ionian Gyre? If so cite it here, with acronym. Line 25. "model data" -> "model output".

1 Introduction

Page 2 Line 2. "satellite based" -> "satellite-based". Need a hyphen when an adjective is made of two or more words. Line 5. "nutrients availability" -> "nutrient availability". Adjective substantive always singular. Line 9. "whole" -> "overall". Line 15. "depress nutricline" -> "depress the nutricline". Line 16. "subtropical (subpolar) gyre"-> Replace

either with "a subtropical (subpolar) gyre" or with "subtropical (subpolar) gyres". Line 17. "which maintains the downwelling (upwelling)". To my knowledge, a stationary gyre, such as the great gyres of the oceans, doesn't cause up- or down-welling (vertical velocity should be zero in a common stationary case). It's a quasi-geostrophic, i.e. time-evolving situation that does. I do understand, though, that nutrients are kept more (less) distant from the surface at the center of an anticyclonic (cyclonic) feature, e.g. because of its depressed (uplifted) pycnocline. Please correct or comment (and provide reference). Lines 18-20. This is correct because upwelling (downwelling) is actually caused when a cyclonic (anticyclonic) gyre FORMS. Line 25. "time-life" -> "lifetime ". Change throughout text. Line 26. "spring" -> "the spring". Line 31. "(NIG, see Figure 1)". The NIG is absent from either Fig. 1 or its caption. Please highlight and describe it clearly. Line 34. "time-scale" -> "time scale". Non need for hyphen, change throughout text.

Page 3 Line 1. "feedback with the Adriatic Sea" -> add "(see below)". Line 3. "the vertical structure... have been" -> "the vertical structure... has been" Line 21. "during cyclonic" -> "during the cyclonic". Also, "downlift" -> "depression" or "lowering". There is no such thing as "lifting down". Change throughout text, please. Line 26. "initiation" -> "start"

2 Data and Methods

2.1 Satellite and modelling data

Page 4 Line 7. "8-day average maps time-series contains" -> "the 8-day average maps time series contains". Again, "map" is singular here, and no hyphen for "time series". Line 8. "80% of data" -> "80% of good data". Same for the 96% sentence. Also "linear" -> "a linear". Line 9. "Then, time-series". -> "Then, the time series". Line 21. "four air-sea fluxes components" -> "four air-sea flux components". Line 23. "as time-integral" -> "as the time-integral". Line 25. "Nucleous" -> "Nucleus". Line 31. "product performances" -> "product performance".

**2.2 In-situ data**

Page 5 Line 1. "from Coriolis" -> "from the Coriolis". Line 11. "nutrient depleted" -> "nutrient-depleted".

**2.3 Phenological metrics**

Line 14. "[Chl-a] time-series" -> "[Chl-a] remotely sensed time series" Line 16. "referred by" -> "referred to as". Line 19. "main growing period" -> "main growth period". Line 21. "trophic situation" -> "the trophic situation". Line 22. "March spatially" -> "March, spatially". Line 23-24. "Date of the" -> "The date of the". Line 24. "determine bloom initiation date although" -> " determine the bloom onset date, although". Needs an article and a comma. Line 25. "methodology... require then" -> "methodology... requires then" Lines 25-26. "8-day annual" -> "The 8-day annual"

**3 Results and Discussion 3.1 Physical and chemical characterization of the NIG**

Line 3. "Time-series of ADT" -> "The time series of ADT" Line 4. "1992-1997." -> ""1992-1997. (Figure 2a)". Please cite figure right away, for the ease of the reader. Line 4 "Anticyclonic" -> "The anticyclonic". Line 6. "circulation was" -> "the circulation was" Line 7. "blooming period" -> "bloom" (suggested: "blooming period" in English sounds somewhat vernacular!) "ADT difference" -> "the ADT difference" Line 8. "Resulting values" -> "The esulting values"; "on Figure 2a" -> "in Figure 2a"; "referred in" -> "referred to in". Line 9. "Although, ICI" -> "Although the ICI". Add "the" and eliminate comma. Line 10. "by seasonal" -> "by the seasonal". Line 12. "small scale" -> "small-scale". Line 13. "2006, circulation" -> "2006, the circulation". Line 14. "On Figure 2b" -> "In Figure 2b". Line 17. "higher ADT value" - > "a higher ADT value". Line 18. "transition from an anticyclonic (Figure 2d)". I have difficulty in attributing an anticyclonic circulation to the ADT pattern of Fig. 2d. Could the Authors better illustrate this circulation? By eye, it doesn't seem conceptually (sign-wise) different from the cyclonic

pattens, though with less negative ADT values in the north. BTW I am OK with Fig. 2b's anticyclonic pattern, but Fig. 2d doesn't look like Fig. 2b. Line 20. "June 2012, NIG" -> "June 2012, the NIG". Line 21. "the period" -> "the periods". Line 22. "Resulting" -> "The resulting". Line 25. "to the cyclonic" -> "to describe the cyclonic". Line 29. "a steep gradient is observed" -> "a steep gradient, implying isopycnal southward deepening, is observed".

Line 5. "when NIG" -> "when the NIG".

3.2 General patterns of phytoplankton phenology in the Ionian Sea

Line 12. "see Figure 4, D'Ortenzio" -> "Figure 4; D'Ortenzio". No need for "see"; "Highest" -> "The highest" Line 15. "lowest" -> "the lowest"; "from October" -> "starting in October"; "Date_GR_Max" -> "The Date_GR_Max". Lines 17-18. "between northern" -> "between the northern". çLine 18. "on the Hovmoeller" -> "in the Hovmoeller". Line 20. "centered on March" -> "centered in March"; "38°N-39°Nband" -> "38°N-39°N band": needs a space. Line 22. "of isopycnal" -> "of the isopycnal"; "during cyclonic" -> "during the cyclonic".

3.3 Impact of the NIG circulation on the [Chl-a] phenology

Line 26. "displays interannual average, over period July 1998 – June 2012" -> more simply: ""displays the July 1998 – June 2012 average". Also, eliminate commas. Line 27. "to interannual" -> "to such interannual". Line 29. "CHL_Year) with" -> better: "CHL_Year), indeed with". Line 31. "only March period is considered" -> "March only is considered"; "CHL_March" -> "the CHL_March"

Line 1. "March which" -> "March, which". Line 4. "In South Ionian" -> "In the South Ionian". Line 5. "by BiOS" -> "by the BiOS". Line 7. "are anticipated by" -> maybe "are early by"? Not sure anticipated is OK. Pls check; "when circulation" -> " "when the

circulation". Line 15. "up to January" -> "to January"; "most of time-series" -> either "most time series" or "most of the time series" (no hyphen). Line 16. "generally few" -> "generally a few". Line 17. "of [Chl-a]" -> "of the [Chl-a]". Line 23. "as it can" -> "as can".

3.4 Role of the NIG circulation compared to the interannual variability in MLD (focus on the region S3)

Section title. I'm not sure about the title construction "Role of... compared to...", a little illogic. Maybe "NIG circulation patterns and MLD variability" or "Role of the NIG circulation in the variability of the MLD" or "The NIG circulation patterns compared to the ... MLD".

Line 27-28. "on the phytoplankton phenology" -> "on phytoplankton phenology". No "the" here. Line 28. "interannual variability also exists", maybe you should add "independently of the BiOS switch"? Do I understand well? Line 29. "to variability of ICI" -> "to the variability of the ICI". Line 31. "affect the bloom development" -> "affect bloom development". (or "the bloom's development") Line 32. "of late [Chl-a] peak" -> "of a late [Chl-a] peak".

Line 1. "were anticipated to" -> "occurred as early as" or "were brought forward to". "to anticipate" means to expect something (check), not to occur early. Lines 2-3. "This...2012". Why don't Authors overplot buoyancy loss anomaly w/ respect to average in Fig. 6 and refer to Fig. 6 in sentences like this one? Once again, words are more cumbersome to digest without a figure. (Add another axis on the left with % difference buoyancy loss). Line 3. "compared to the average for the period 1999-2012" -> "compared to the reference 1999-2012 average". Line 4. "shallow winter MLD"-> "a shallow winter MLD"; "nutrients inputs" -> "nutrient inputs" Again, adjective-substantive always singular, even if nutrients are more than one type, in this case. Line 4. "to surface layer" -> "to the surface layer"; "consequently the spring bloom"->"consequently

hindered the spring bloom". Line 7. "spearman" -> "Spearman". Line 8. "to circulation regime" -> "to the circulation regime". Line 9. "of the [Chl-a]." -> "of [Chl-a]." No "the". Line 10. "is observed, only the year" -> "is observed: only the year". Line 11. "a ICI" -> "an ICI". Line 13. "circulation pattern" -> "the circulation pattern". Line 14-15. "when circulation is cyclonic and apparently affect" -> "when the circulation is cyclonic and apparently affects". Line 19. "uplift/downlift" -> "uplift/depression"; "cyclonic circulation" -> "the cyclonic circulation". Line 26. "overpasses" -> "overtakes in depth" or "lies deeper than". Line 31. "enlighten" -> "illuminated". Line 32. "in trades biome" -> "in the Trades' biome"

Line 3. "wind driven" -> "wind-driven". Line 4. "that MLD" -> "that the MLD". Line 8. "nutrients supplies" -> "nutrient supplies". Line 9. "other not." -> "other not receiving such supply." Line 12. "Nitracline" -> "The nitracline"; "with difference density" -> "with the difference density"; "mg m-3" -> "kg m-3". Same corrections for the 0.03 case (add "the" and change to "kg m-3"). Line 15. "surface indicating" -> "surface, indicating". Line 17. "up to" -> "down to". Line 19. "a entrainment bloom (nutrient limited)" -> "an entrainment bloom (nutrient-limited)". What is an entrainment bloom? LIne 20. "light limited" -> "light-limited"; "initiation" -> "onset". Line 22. "shallowing" -> "shoaling". Line 23. "that phytoplankton bloom" -> "that a phytoplankton bloom" or " that phytoplankton blooms". Line 29. "MLD" -> "the MLD".

Line 2. "light driven" -> "light-driven". Line 4. "make us conclude" -> "induce us to conclude". Line 5. " nutrient driven" -> "nutrient-driven". Line 6. " [Chl-a)" -> " [Chl-a]".

4 Summary and Conclusion -> 4 Summary and Conclusions

Line 10. "supply in the surface layer" -> "supply to the surface layer". Lines 10-11. "high all along the year" -> "high year-round". Line 13. "sustains phytoplankton bloom"

-> "sustains the phytoplankton bloom". Line 19. "and winter deepest MLD" -> "and the winter deepest MLD". Line 25. "large amount" -> "a large amount".

Figures and captions

Figure 2 caption. "(S1 – S2, see Figure 1c)" ->"(S1 – S2, see Figure 1b)". Fig. 3 caption. Maybe add "black dots indicate in situ stations used for the maps". Figure 4 caption. "satellite [Chl-a]" -> "8-day satellite [Chl-a]". Remind reader of temporal resolution. Figure 5. Even though you have units spec'd in the caption, I suggest you add the units on top of the palettes, i.e. mg m-3, month (this not strictly necessary) and %. Always for the ease of the reader. It can be done quickly, e.g. w/ Powerpoint. Fig 6. Characters are a bit small, in the Fig. Please enlarge (in view of drastic figure reduction by editorial process). Also, please add units on axes. Figure 7. Again, characters are small and isopycnal line almost invisible. Please enlarge chars, and thicken and change color to line.

---

## Referee Comment (RC2) · Anonymous Referee #2 · 26 Feb 2018

Summary:

This manuscript contributes to improve the studies about the relationships between the phytoplankton phenology and the cyclonic/anticyclonic circulation, performed in the North Ionian Sea. This paper is of scientific relevance, well written and logically organized. They present several tables and figures including lot information published about the issue. Unfortunately, however the results presented are not yet fully convincing in its present form, and some further work is needed to provide more solid evidence to some of the -at present- speculative results shown.

General comments:
Language and grammar: generally, the manuscript is well written. Just a few sentences would need to be revised; there are some typos that would need to be corrected. Title: The title reflects most of the authors guidelines in the manuscript.

Abstract: The abstract presents a good summary of the manuscript. The context of the study is clearly defined. A suggestion could be to highlight the obtained results better.

Introduction: well written and exhaustive. However, in Page 3 - Line 15-16, authors should rewrite these sentences. Several authors of Mayot et al (2016) study coincide with the current work. The authors indicate that Mayot et al don't analyze the interannual variability, when the title itself is "Interannual variability of the ....". The authors should rewrite these sentences and specify what improvement or advance present this study with respect to the study of Mayot et al. (2016). Technical comment: Page 3, Line 13: Define CZCS and SeaWIFS.

Material and Methods: One of the main concerns I detect in the study is that they use chlorophyll concentration obtained from OC-CCI database that has been built using standard algorithms. It is known by all that in the Mediterranean, these algorithms do not work correctly and there are several studies, including some made by the same authors of this work, which indicate the use of specific chl-a algorithms for the Mediterranean Sea. Authors should use these algorithms (MedOC4, Volpe et al. 2007 for Case 1 waters and the AD4 algorithm for Case 2 waters type D'Alimonte and Zibordi, 2003) or demonstrate, through a statistical analysis, that there are no differences in the results of this study using standard instead of regional algorithms.

D'Alimonte D. and Zibordi G. (2003). Phytoplankton Determination in an Optically Complex Coastal Region Using a Multilayer Perceptron Neural Network. IEEE Trans. Geosci. Remote Sensing, vol. 41, pp. 286.

Volpe, G., Santoleri, R., Vellucci, V., Ribera d Acalà, M., Marullo, S., and D Ortenzio, F. (2007). The colour of the Mediterranean Sea: Global versus regional bio-optical algorithms evaluation and implication for satellite chlorophyll estimates. Remote Sens.

BGD
Environ., 107, 625-638.

On the other hand, I suggest that the authors can work with the Marine Copernicus database (OCEANCOLOUR\_MED\_CHL\_L3\_REP\_OBSERVATIONS\_009\_073) that uses regional chlorophyll algorithms as well as a better spatial resolution (1km x 1km).

In fact, the authors use this type of data for temperature and salinity (Page 4, line 25). Results and discussion: in general, all the topics are well-treated and detailed. Anyway, several suggestions are given in the "technical comments" with the intent to improve the comprehension of the text.

Page 5, Lines 8-9. Why the authors use the value of 1uM to establish the nitracline? In addition, the authors should explain why they only work with nitrate and not with phosphate, when it is known by the scientific community to be a very important nutrient in the Mediterranean Sea.

Page 7, Lines 1-8, The authors found a linear relationship between the depth of the nitracline and the 28.9 kg/m3 isopycnal. The authors should clarify if this relationship is for the whole year or only for certain dates, since it can be modified clearly depending on the time (mixed or stratification period). This issue is very important since then you use this relationship to build figure 7.

Page 9, Line 25-30. The authors should provide empirical data to support these hypotheses.

Figures and tables: Figures and tables are appropriate. Anyway, minor suggestions are given below:

- Figure 4. Please, include in Figure 1 the line that has been performed the Hovmöeller diagram. - Write properly "Hovmöeller" throughout the text

References: prior work are fully cited

---

## Author Comment (AC1) · 25 Mar 2018

Response to anonymous Referee #1.

We would like to thank the referee #1 for its review and the interest he shows to our manuscript. We are also very grateful for the careful review of the English. Regarding linguistic corrections, we changed the manuscript according to his suggestions. Changes appear in red in the new version of the manuscript. In the following we answer the comments he had about science.

(1) To my knowledge, a stationary gyre, such as the great gyres of the oceans, doesn't

cause up- or down-welling (vertical velocity should be zero in a common stationary case). It's a quasi-geostrophic, i.e. time-evolving situation that does. I do understand, though, that nutrients are kept more (less) distant from the surface at the center of an anticyclonic (cyclonic) feature, e.g. because of its depressed (uplifted) pycnocline. Please correct or comment (and provide reference).

Authors response: We agree that in basin scale quasi-geostrophic gyre, vertical velocity is null so there are no nutrient supplies by this way. We wanted to refer to the impact of cyclonic and anticyclonic circulation on the position (depth) of the nutricline. To avoid confusion, we removed the mention to downwelling/upwelling line 16 (page 2) and changed the text accordingly:

"At basin scale, the trophic status of subtropical (subpolar) gyres is determined by anticyclonic (cyclonic) circulation, which controls the depth of the nutricline. " (page 2, lines 16-17)

(2) I have difficulty in attributing an anticyclonic circulation to the ADT pattern of Fig. 2d. Could the Authors better illustrate this circulation? By eye, it doesn't seem conceptually (sign-wise) different from the cyclonic pattens, though with less negative ADT values in the north. BTW I am OK with Fig. 2b's anticyclonic pattern, but Fig. 2d doesn't look like Fig. 2b. Line 20.

Authors response: We attributed an anticyclonic circulation pattern to the figure 2d because one can observe a tongue of higher ADT values (about –2 cm) expending from the Mid Ionian Jet to the north. A small yellow spot is visible around 38°N in the center of the North Ionian Jet. Although it is less clear than for a typical anticyclonic pattern (Figure 2C and not 2b), a similar structure is observed in Figure 2d. Please note that Figure 2b represents the cyclonic situation.

The following sentence has been added to the text (page 6, line 13)

"Indeed, similarly to Figure 2c, in Figure 2d a tongue with higher ADT values extents

northward from the Mid Ionian Jet to the center of the NIG."

(3) Section title. I'm not sure about the title construction "Role of... compared to...", a little illogic. Maybe "NIG circulation patterns and MLD variability" or "Role of the NIG circulation in the variability of the MLD" or "The NIG circulation patterns compared to the ... MLD".

Authors response: Title of the section has been changed to: "Role of the NIG circulation patterns compared to the interannual variability in MLD (focus on the region S3)"

(4) Why don't Authors overplot buoyancy loss anomaly w/ respect to average in Fig. 6 and refer to Fig. 6 in sentences like this one? Once again, words are more cumbersome to digest without a figure. (Add another axis on the left with % difference buoyancy loss).

Authors response: We agree that figures are better than words. However, we can't overplot Buoyancy loss anomaly on Figure 6 because it is not a time-series but an annual value per year (it is winter integrated). In addition, Figure 6 is already relatively dense with many time-series in the same figure panel, that is why we estimate that to give winter buoyancy loss values in a table is the best option for clarity.

(5) What is an entrainment bloom? Line 20.

Authors response: An entrainment bloom as defined by Cullen (2002) is a bloom driven by the deepening of the MLD in case of nutrient limitation. To be clearer the following definition has been added line 15 page 10.

"A common explanation for the fall and spring bloom is the succession of an entrainment bloom (i.e. bloom driven by MLD deepening in nutrient-limited conditions) in fall followed by a spring bloom (light-limited, Cullen et al. 2002; Levy et al., 2005)."

(6) Figures:

Figure 1 add the position of the NIG

Authors response: Thank you for your suggestion, it has been added.

Figure 2 caption. "(S1 – S2, see Figure 1c)" ->"(S1 – S2, see Figure 1b)".

Authors response: Thank you noticing this mistake. It has been corrected in the new version of the manuscript.

Fig. 3 caption. Maybe add "black dots indicate in situ stations used for the maps".

Authors response: Caption of Figure 3 has been modified accordingly:

"Climatological maps of the estimation of the depth of the isopycnal 28.9 kg m-3 for cyclonic (panel a) and anticyclonic (panel b) regimes. Data were spatially interpolated with kriging, black dots indicate in situ data sued for kriging. Relationship between the depth of the nitracline and the depth of isopynal 28.9 kg m-3 in the North Ionian Sea (panel c) from in situ data collected during three oceanographic cruises (see colors). "

Figure 5. Even though you have units spec'd in the caption, I suggest you add the units on top of the palettes, i.e. mg m-3, month (this not strictly necessary) and %. Always for the ease of the reader. It can be done quickly, e.g. w/ Powerpoint.

Authors response: Thank you for your suggestion, units have been added on the top of palettes.

Fig 6. Characters are a bit small, in the Fig. Please enlarge (in view of drastic figure reduction by editorial process). Also, please add units on axes.

Authors response: Thank you for your suggestion, we enlarged all characters in the new version of the manuscript.

Figure 7. Again, characters are small and isopycnal line almost invisible. Please enlarge chars, andthicken and change color to line

Authors response: Thank you for your suggestion. It has been done in the new version of the manuscript

---

## Author Comment (AC2) · 25 Mar 2018

We would like to thank the referee #2 for its review and the interest he shows to our manuscript. In the following we answer the comments one by one.

(1) In the abstract: A suggestion could be to highlight the obtained results better.

Author response: Thank you for your suggestion, the following sentences have been added to the abstract

"Two trophic regimes were then identified in the NIG and they could be explained with the relative position of the MLD and nitracline. The first one is characterised by an

early winter bloom onset and the absence of chlorophyll peak in March. It was observed when circulation was anticyclonic or when winter MLD was relatively shallow. A dominant regenerated production all year round and an absence of significant nutrient supplies to surface waters are proposed to explain this trophic regime. Conversely, the second trophic regime is marked by a bloom onset in late winter (i.e. February) and a peak of chlorophyll in March. The chlorophyll increase was interpreted as the direct response to nutrient enrichment of surface waters. This winter/spring bloom was observed when circulation was cyclonic and when winter mixing was relatively strong.

(2) Introduction: In Page 3 - Line 15-16, authors should rewrite these sentences. Several authors of Mayot et al (2016) study coincide with the current work. The authors indicate that Mayot et al don't analyze the inter-annual variability, when the title itself is "Interannual variability of the ...". The authors should rewrite these sentences and specify what improvement or advance present this study with respect to the study of Mayot et al. (2016).

Athors response: We fully agree that Mayot et al. (2016) study interannual variability in phenology and that is what we implied in sentence line 15 : "More recently, Mayot et al. (2016) showed that phytoplankton phenology is extremely dynamic in the North Ionian Sea with significant changes in the annual cycle of surface chlorophyll-a concentration from year to year."

The improvement of the paper compared to the one of Mayot et al. (2016) is that it focuses on the causes of change in phytoplankton phenology which is out of the scope of the Mayot et al. (2016) paper. In particular, the impact of the decadal changes in BiOS circulation (switching from cyclonic to anticyclonic and vice-versa) on phytoplankton phenology is investigated in our paper.

To avoid confusion lines 15 and 16 have been modified to:

"More recently, Mayot et al. (2016) studied interannual variability in phytoplankton phe-
nology in the Mediterranean Sea. Authors showed that phytoplankton phenology is extremely dynamic in the North Ionian Sea with significant changes in the annual cycle of surface chlorophyll-a concentration from year to year. However, they did not investigate the potential causes of this interannual variability. "

(3) Page 3, Line 13: Define CZCS and SeaWIFS.

Authors response: This is done in the new version of the manuscript. Thank you for the suggestion.

(4) Material and Methods: One of the main concerns I detect in the study is that they use chlorophyll concentration obtained from OC-CCI database that has been built using standard algorithms. It is known by all that in the Mediterranean, these algorithms do not work correctly and there are several studies, including some made by the same authors of this work, which indicate the use of specific chl-a algorithms for the Mediterranean Sea. Authors should use these algorithms (MedOC4, Volpe et al. 2007 for Case 1 waters and the AD4 algorithm for Case 2 waters type D'Alimonte and Zibordi,2003) or demonstrate, through a statistical analysis, that there are no differences in the results of this study using standard instead of regional algorithms.

D'Alimonte D. and Zibordi G. (2003). Phytoplankton Determination in an Optically Complex Coastal Region Using a Multilayer Perceptron Neural Network. IEEE Trans. Geosci. Remote Sensing, vol. 41, pp. 286.

Volpe, G., Santoleri, R., Vellucci, V., Ribera d Acalà, M., Marullo, S., and D Ortenzio, F. (2007). The colour of the Mediterranean Sea: Global versus regional bio-optical algorithms evaluation and implication for satellite chlorophyll estimates. Remote Sens.

On the other hand, I suggest that the authors can work with the Marine Copernicus database (OCEANCOLOUR\_MED\_CHL\_L3\_REP\_OBSERVATIONS\_009\_073) that uses regional chlorophyll algorithms as well as a better spatial resolution (1km x 1km). BGD
In fact, the authors use this type of data for temperature and salinity (Page 4, line 25).

Authors response: we agree that global Chl-a products are not the most adapted to retrieve accurate estimations of Chl-a in the Mediteranean Sea as they tend to overestimate Chl-a concentration in oligotrophic conditions (Chl-a

Mayot et al., 2016) which are also based on global [Chl-a] products."

Bricaud, A., Bosc, E., & Antoine, D. (2002). Algal biomass and sea surface temperature in the Mediterranean Basin: Intercomparison of data from various satellite sensors, and implications for primary production estimates. Remote Sensing of Environment, 81(2-3), 163-178.

O'Reilly, J.E., Maritorena, S., Siegel, D.A., O'Brien, M.C., Toole, D., Mitchell, B.G., Kahru, M., Chavez, F.P., Strutton, P., Cota, G.F. and Hooker, S.B., 2000. Ocean color chlorophyll a algorithms for SeaWiFS, OC2, and OC4: Version 4. SeaWiFS postlaunch calibration and validation analyses, Part, 3, pp.9-23.

D'Ortenzio, F. and Ribera d'Alcalà, M., 2009. On the trophic regimes of the Mediterranean Sea: a satellite analysis. Biogeosciences, 6(2), pp.139-148.

(5) Page 5, Lines 8-9. Why the authors use the value of 1uM to establish the nitracline? In addition, the authors should explain why they only work with nitrate and not with phosphate, when it is known by the scientific community to be a very important nutrient in the Mediterranean Sea.

Author response: Isoline  $1\mu$ M is commonly used to estimate of the nitracline depth which separates surface nitrate depleted waters to deep nitrate repleted waters, in the global ocean (Cermeno et al., 2008) and in the Mediterranean Sea (Lavigne et al., 2013; Pasqueron de Fommervault et al. 2015). In practice, we found also this measure more robust than depth of maximum gradient to estimate nitracline.

This is true that Phosphate is considered the most limitant element regarding phytoplankton growth in the eastern Mediterranean Sea as N/P ratio is higher than the Redfield ratio. However, experiments have shown that phytoplankton growth is co-limited by nitrate and phosphate in the Mediterranean Sea (Psarra et al., 2005; Thingstad et al., 2005). As the quality of phosphate data in databases are poorer than the quality of nitrate data (because of the very small phosphate concentrations) we preferred
to work with nitrate data. This choice is not unusual as previous Mediterranean Sea works were also based on nitrate (D'Ortenzio et al., 2014; Lavigne et al., 2013).

The following sentences have been added page 5, lines 13:

"Although it was demonstrated that phytoplankton growth is co-limited by nitrate and phosphate in the Eastern Mediterranean Sea (Psarra et al., 2005; Thingstal et al., 2005), only nitrate dynamic is considered here as the both elements co-varies and control phytoplankton growth (Ribera d'Alcalà et al. 2003) and as the quality of phosphate data in database is not good enough for such analysis. "

Cermeño, P., Dutkiewicz, S., Harris, R.P., Follows, M., Schofield, O. and Falkowski, P.G: The role of nutricline depth in regulating the ocean carbon cycle. Proceedings of the National Academy of Sciences, 105(51), pp.20344-20349, 2008.

Psarra, S., Zohary, T., Krom, M.D., Mantoura, R.F.C., Polychronaki, T., Stambler, N., Tanaka, T., Tselepides, A. and Thingstad, T.F.: Phytoplankton response to a Lagrangian phosphate addition in the Levantine Sea (Eastern Mediterranean). Deep Sea Research Part II: Topical Studies in Oceanography, 52(22-23), pp.2944-2960, 2005.

Thingstad, T.F., Krom, M.D., Mantoura, R.F.C., Flaten, G.F., Groom, S., Herut, B., Kress, N., Law, C.S., Pasternak, A., Pitta, P. and Psarra, S.: Nature of phosphorus limitation in the ultraoligotrophic eastern Mediterranean. Science, 309(5737), pp.1068-1071, 2005.

Lavigne, H., D'Ortenzio, F., Migon, C., Claustre, H., Testor, P., d'Alcalà, M. R., Lavezza, R., Houpert, L. and Prieur, L.: Enhancing the comprehension of mixed layer depth control on the Mediterranean phytoplankton phenology, Journal of Geophysical Research: Oceans, 118, 3416–3430, doi: 10.1002/jgrc.20251, 2013.

Pasqueron de Fommervault, O., D'Ortenzio, F., Mangin, A., Serra, R., Migon, C., Claustre, H., Lavigne, H., Ribera d'Alcalà, M., Prieur, L., Taillandier, V., Schmechtig, C., Poteau, A., Leymarie, E., BessonF., and Obolensky, G.: Seasonal variability of nutrient

BGD
concentrations in the Mediterranean Sea: contribution of Bio-Argo floats. Journal of Geophysical Research, 120, doi:10.1002/2015JC011103, 2015.

D'Ortenzio, F., et al.: Observing mixed layer depth, nitrate and chlorophyll concentrations in the Northwestern Mediterranean: A combined satellite and no3 profiling floats experiment, Geophysical Research Letters, 2014.

(6) Page 7, Lines 1-8, The authors found a linear relationship between the depth of the nitracline and the 28.9 kg/m3 isopycnal. The authors should clarify if this relationship is for the whole year or only for certain dates, since it can be modified clearly depending on the time (mixed or stratification period). This issue is very important since then you use this relationship to build figure 7.

Author response: The linear relationship has been built with data collected over different seasons and years but always avoiding the winter mixing period (i.e. October 1991; April-May 1992 and April-May 1999). We agree that the relationship is not anymore valid if winter mixing reaches the nitracline. This is illustrated on figure 7 when nitracline is observed at the surface after a deep mixing event. In the new version of the manuscript we better warn the reader about the limits of this relationship: page 7 line 8 the following sentences has been added: "Data used here were collected at different seasons (i.e. in October 1991 and April/May 1992 and 1999) but always during the long period during which water column is stratified. As soon as vertical mixing reaches nitracline depth, this relationship is not anymore valid. Nevertheless, such events are rare and short in the North Ionian Sea."

(7) Page 9, Line 25-30. The authors should provide empirical data to support these hypotheses.

Authors response: We attempted to support these hypotheses with model data presented in Figure 7. In situ data presented in Figure 3 were not sufficient to analyse interannual variability.
(8) Figure 4. Please, include in Figure 1 the line that has been performed the Hov-möeller diagram.

Authors response: Thank you for your suggestion, the line has been added in the new version of the figure 1.

(9) Write properly "Hovmöeller" throughout the text

Authors response: Corrections have been done. Thank you.
Fig. 1.
Comparison of the phenological metrics obtained year 2004-2003 with (1) OC-CCI product (0C4v6 algorithm for Chl-a) and (2) Volpe et al., (2007) algorithm.
Fig. 2.

---

## Author Response (AR2)

Dear Dr. Gregoire,

We would like to thank you and the referees for the interest you showed to our manuscript and the careful review you did.

We have modified the manuscript to integrate all of your corrections as well as the ones of reviewer #1. All the new modifications appeared in blue in the manuscript and the figure 3 has been modified to integrate the unit at the color scale.

Looking forward to hearing from you.

Best regards.

Héloïse Lavigne and co-authors.
* * *
List of modifications:

Editor:

Minor comments:
Abstract: please define NIG : Done
Page 5, line 25 : « as both elements … » changed to "as the both elements co-vary"
Page 6, line 15: difference between:  the space has been added between the two words
Page 7, line 3: be attributed:  the space has been added between the two words
Page 7, line 25, nutrients to: the space has been added between the two words
Figure 3, legend, « used »: It has been corrected
Page 8, line 31, During:  It has been corrected

Reviewer:

Abstract

Line 20. "gyre, bloom onset occurred" -> "gyre, the bloom started". Done
Line 29. "peak of chlorophyll" -> "chlorophyll peak". Done

1 Introduction

Line 10. "Mixed Layer Depth (MLD)" -> "The Mixed Layer Depth (MLD)". Done
Line 24. "the formation of cyclonic eddies cause" -> " the formation of cyclonic eddies causes". Done
Line 30. "long lifetime" -> " long-lived". Done
Line 31. "superior" -> "greater than". Done

2 Data and Methods

2.1 Satellite and modelling data

Line 9. "8-day average". -> "the 8-day average". Done
Line 15. "as well as the OC4 global algorithm" -> "as for OC4 ". Done
Line 17. "As the present study focus" -> "As the present study focuses". Or "focusses", if British English is needed. Modified to "As the present study focuses"
Line 18. "utilisation of global" -> "utilisation of the global". Done
Line 30. "(ECMFW)" -> "(ECMWF)". Done
Line 31. "Data was available" -> "Data were available" Done

2.2 In-situ data

Line 5. "isopycnal 28.9 kg m-3" -> "28.9 kg m-3 isopycnal" (correct twice in line) Done
Line 25. "nitrate dynamic" -> "nitrate dynamics" Done
Line 25. "as the both elements co-varies" -> "since both nutrients co-vary" Done
Line 25. "and as the quality" -> "since the quality" Done

2.3 Phenological metrics

Line 4. "in the Mediterranean Sea trophic situation" -> "the Mediterranean Sea trophic situation". Done
Line 6. "initiation" -> "onset". Done
Line 7. "determine bloom onset date" -> " determine the bloom onset date". Done

3 Results and Discussion
3.1 Physical and chemical characterization of the NIG

Line 15. "differencebetween" -> "difference between" Done
Line 21. "Although, the ICI" -> "Although the ICI": eliminate comma. Done
Line 22.
Line 22. "it is affected by seasonal signal" -> "the differences with respect to the yearly means suggest that it is affected by a seasonal signal". (suggested correction) Done
Line 27-28 "(see Figure 2c)" -> "(Figure 2c)". No need for "see". Usually one should use "see" only to suggest consultation of "remote" figures w/respect to the text (e.g. in the Introduction: "see Fig. 4 below"). OK, "see" has been removed.

Line 3. "beattributed" -> "be attributed". Done
Line 16. "Haine 2010" -> "Haine, 2010" Done

Line 19. "Data used here" -> "The data used here" Done
Line 20. "during which water column" -> "during which the water column" Done
Same. "reaches nitracline depth" -> "reaches the nitracline depth" Done
Line 21. "Applying" -> "By applying" Done
Line 25. "nutrientsto" -> "nutrients to" Done

3.2 General patterns of phytoplankton phenology in the Ionian Sea

Line 33. "then," -> eliminate comma Done

Line 2. "(see Figure 5)" -> "(Figure 5)" Done

3.3 Impact of the NIG circulation on the [Chl-a] phenology

Line 15. "negative anomalies" -> more simply: " negative anomalies (Fig. 5c)". Cite figures. Done
Line 16. "panels" -> "Figure 5". c
Line 30. "end of January" -> "end of January (Figure 6, right panel)" Done
Line 31. "D uring" -> "During" and "the cyclonic regime," -> "the cyclonic regime (Figure 6, left panel)" Done
Line 33. "The annual maximum growth rate..." -> cite Figure 7 here? Reference to Table 1 has been added.

Line 1. "in average" -> "on average" Done

3.4 Role of the NIG circulation patterns compared to the interannual variability in MLD (focus on the region S3)

Line 14. "on the area S3" -> "on the S3 area" Done
Line 20. "as earlier as" -> "as early as" Done
Line 21. "weak:-24%" -> "weak: -24%" insert space after column. Done
Line 21-22. "(see Table 1)" -> "(Table 1)". Done

Line 17. "Trades' " -> "the Trades' " Done
Line 23. "very small" -> "very low" Done
Line 28. "such supply" -> " such a supply" Done
Line 30. "estimated" -> "the estimated" Done

Line 2. "reached almost" -> "almost reached" Done
Line 4. "a entrainment" -> "an entrainment" Done
Line 25. "masksthe" -> "masks the" Done

4 Summary and Conclusion -> 4 Summary and Conclusions (CORRECT THIS unless the journal wants "Conclusion")

Line 28. "of spring bloom" -> "of a spring bloom". Done

Lines 2-3. "in case of anticyclonic circulation" -> "in the anticyclonic case" Done
Line 12. "high dense" -> "high-density" or "very dense" Done
Line 22. "DhyanAranha" -> "Dhyan Aranha" (two words, I guess?) Yes, it has been modified.
Line 25. "Equipementd'Avenir" -> " Equipement d'Avenir " Done
Line 27. "are thanks" -> "are thanked" Done

Figures and captions

Fig. 3 a, b color palette: units are missing. I suppose they are meters? Maybe add in caption or put an "m" near the palette. The unit "m" has been added to the Figure.
Figure 6 caption. "regimesaveraged in the region S3" -> "regimes averaged in region S3" Done